# Heterochromatin protein 1a functions for piRNA biogenesis predominantly from pericentric and telomeric regions in *Drosophila*

Ryan Yee Wei Teo[1,2,5], Amit Anand[1], Vishweshwaren Sridhar[1], Katsutomo Okamura [1,3] & Toshie Kai[4]

In metazoan germline, Piwi-interacting RNAs (piRNAs) provide defence against transposons. Piwi–piRNA complex mediates transcriptional silencing of transposons in nucleus. Heterochromatin protein 1a (HP1a) has been proposed to function downstream of Piwi-piRNA complex in *Drosophila*. Here we show that *HP1a* germline knockdown (HP1a-GLKD) leads to a reduction in the total and Piwi-bound piRNAs mapping to clusters and transposons insertions, predominantly in the regions close to telomeres and centromeres, resulting in derepression of a limited number of transposons from these regions. In addition, *HP1a*-GLKD increases the splicing of transcripts arising from clusters in above regions, suggesting HP1a also functions upstream to piRNA processing. Evolutionarily old transposons enriched in the pericentric regions exhibit significant loss in piRNAs targeting these transposons upon *HP1a*-GLKD. Our study suggests that HP1a functions to repress transposons in a chromosomal compartmentalised manner.

[1] Temasek Life Sciences Laboratory, 1 Research Link, National University of Singapore, 117604 Singapore, Singapore. [2] Department of Biological Sciences, National University of Singapore, 117543 Singapore, Singapore. [3] School of Biological Sciences, Nanyang Technological University, 60 Nanyang Drive, 637551 Singapore, Singapore. [4] Graduate School of Frontier Biosciences, Osaka University, Suita, Osaka 565-0871, Japan. [5] Present address: Department of Pathology, Tan Tock Seng Hospital, 11 Jalan Tan Tock Seng, Singapore 308433, Singapore. These authors contributed equally: Ryan Yee Wei Teo, Amit Anand. Correspondence and requests for materials should be addressed to A.A. (email: amit@tll.org.sg) or to T.K. (email: toshie_kai@fbs.osaka-u.ac.jp)

Transposons constitute a large portion of genome in the majority of eukaryotes, and their mobilisation in germline poses a threat to fecundity and fitness of the population[1,2]. Animals, plants, and fungi have evolved RNA-based systems for transcriptional and post-transcriptional transposon silencing[3]. In the animal germline cells, a specialised small RNA-based defence system, termed Piwi-interacting RNA (piRNA) pathway, is primarily responsible for transposon silencing[4].

Most proteins involved in piRNA pathway are conserved[4]. *Drosophila* has been a very useful model system for studying the mechanistic details of the piRNA pathway. The piRNAs are processed from longer precursors transcribed from genomic regions containing clusters of fragmented transposons known as piRNA clusters[5]. These are transcribed in a non-canonical manner, which requires the RDC complex comprising Rhino (Rhi), Deadlock (Del), and Cutoff (Cuff)[6–9]. Rhi and cuff were reported to repress the precursor splicing, required for a proper piRNA processing[6, 9]. In germline cells, piRNAs are processed by primary and secondary processing mechanisms. Primary processing generates piRNAs that target transposons from a single-stranded precursor RNAs. Secondary processing involves a primary piRNA-guided cleavage of transposon sense transcripts to produce sense piRNAs, which then guide the cleavage of cluster transcripts for antisense piRNA production, forming a piRNA amplification loop, termed ping-pong cycle[4]. The piRNAs are loaded onto Piwi, the founding member of PIWI-clade proteins, and translocate into the nucleus. The Piwi–piRNA complex, also referred as the piRNA-induced silencing complex (Piwi–piRISC), silences transposons transcriptionally by inducing heterochromatin formation[8,10–15].

Piwi has been proposed to function with downstream partners, but very little information about the partners is available. For instance, a nuclear protein, Panoramix (Panx) mediates Piwi-piRISC-directed recruitment of dSETDB1/Egg, a histone methyltransferase, at the transposon loci for H3K9 trimethylation[14,15]. Another reported Piwi partner, the conserved heterochromatin protein, Heterochromatin Protein 1a (HP1a) has been proposed as a downstream factor to enforce transposon silencing in the *Drosophila* germline and ovarian somatic cells[13,16]. Recently, HP1a binding was shown to lead to the repression of transposon loci, which is most likely based on its ability to recruit the Egg protein[14,17–19]. This suggests that HP1a may play its roles downstream of the piRNA pathway. However, the effects of HP1a depletion from the germline on the piRNAs and piRNA pathway proteins have not been studied previously. Therefore, we lack a clear understanding of the HP1a functions associated with the piRNA pathway.

In this study, we examined the function of HP1a in the piRNA pathway and transposon repression in the *Drosophila* female germline cells. Our results suggest that HP1a is required for piRNA biogenesis predominantly from the regions close to centromeres and telomeres. We also show that HP1a functions upstream to piRNA processing, likely by repressing splicing of piRNA precursors.

## Results

**HP1a is required for repression of a subset of transposons.** We depleted HP1a in the *Drosophila* female germline by expressing short hairpin RNA (shRNA) from the Transgenic RNAi Project (TRiP) lines, using a strong germline driver containing two germline-specific Gal4, *NGT40* and *nos*Gal4. We used two TRiP lines, referred hereafter as RNAiHP1a[2] and RNAiHP1a[3], in which shRNA construct is inserted on chromosomes 2 and 3, respectively[20]. *HP1a* germline knockdown (HP1a-GLKD) with either RNAi line efficiently depleted HP1a expression in germline cells, and caused female sterility (Fig. 1a). Upon HP1a-GLKD with RNAiHP1a[2] (HP1aGLKD[2]), the ovaries appeared morphologically similar to the wild-type ones, while the knockdown with RNAiHP1a[3] (HP1a-GLKD[3]) resulted in somewhat atrophic ovaries (Supplementary Fig. 1a).

A previous study reported transposon derepression upon HP1a germline knockdown using a different RNAi line from Vienna *Drosophila* RNAi collection (VDRC)[13]. Consistently, we observed derepression of a telomeric transposon, *HeT-A*, and its accumulation in the oocytes (Fig. 1a and Supplementary Fig. 1b). To examine the global impact of HP1a-GLKD on transposon repression, we performed deep sequencing analysis of total RNA from the HP1a-GLKD[2] and control ovaries. Of 95 canonical transposon families examined, 23 showed more than 2.5-fold upregulation in the HP1a-GLKD ovaries (Fig. 1b and Supplementary Data 2). Of these, derepression of the telomeric transposons, *HeT-A*, *TART*, and *TAHRE*, was considerably higher than that of the other transposon families (Fig. 1b, c; median derepression: 64-fold vs. 3-fold, respectively). However, many transposons that were reported to be derepressed in other piRNA pathway mutants, such as *I-element*, *blood*, *burdock*, *Rt1b* and *roo* were not significantly upregulated in HP1a-GLKD ovaries (Fig. 1b and Supplementary Data 2)[5,21–25]. We confirmed the selective derepression of the transposons in the HP1a-GLKD replicates using quantitative (q)RT-PCR analysis (Supplementary Fig. 1c). To exclude the possibility that partial transposon derepression is caused by incomplete HP1a depletion or off-target effects, we performed clonal analysis using the *hp1a* null mutant allele, *hp1a05*[26]. Consistently, the egg chambers lacking HP1a expression in the germline clearly accumulated HeT-A but not I-element, further indicating that HP1a is required for the repression of a subset of transposons (Fig. 1d and Supplementary Fig. 1e).

**HP1a-GLKD causes loss of pericentric and telomeric piRNAs.** Transposons are repressed in the gonads via piRNA pathway[5,27,28]. Derepression of selective transposons in the HP1a-GLKD ovaries prompted us to investigate the effect of HP1a-GLKD on piRNAs. We sequenced 18~30-nt small RNAs from HP1a-GLKD[2], HP1a-GLKD[3], and the control green fluorescent protein (GFP)-GLKD ovaries (see Supplementary Data 1 for details). Since HP1a was previously shown to participate in piRNA pathway, likely in association with Piwi[10,12,13,19,29,30], we also sequenced Piwi-bound piRNAs in the HP1a-GLKD ovaries. The HP1a-GLKD resulted in 20%–30% reduction in the canonical transposon-mapping total piRNAs, while no significant change was observed for overall Piwi-bound canonical transposon-mapping piRNAs (Supplementary Fig. 2a–c and Supplementary Data 3). Among over 100 transposon families examined, only 21–26 of them had over two-fold reduction in total piRNAs mapping to them in HP1a-GLKD ovaries with two different RNAi lines (Supplementary Fig. 2d and Supplementary Data 3). Among these, 19 transposon families consistently showed significant reduction in Piwi-bound piRNAs mapping to them in HP1a-GLKD ovaries (Supplementary Fig. 2e; Supplementary Data 3).

Consistent with the transposon derepression pattern, both total and Piwi-bound piRNAs targeting HeT-A were most severely reduced in HP1a-GLKD ovaries (Fig. 2a, b, Supplementary Fig. 2f, and Supplementary Data 3). As an exception to the trend observed for other derepressed transposons, upregulation of *accord*, *tirant*, and *transib1*, was accompanied by the loss of the Piwi-bound piRNAs, but the levels of total cellular piRNAs targeting these transposons were not reduced (Supplementary Fig. 2g). Reduction in piRNAs mapping only to a limited number

of transposon families in HP1a-GLKD ovaries was in contrast to the more wide-spread piRNA reduction in the mutants of piRNA pathway components required for the global piRNA biogenesis[4,22,25,29,31,32].

We next examined effect of HP1a-GLKD on the cluster-derived piRNAs. We observed 45%–56% reduction in the total cluster-mapping piRNAs in two RNAi lines, and approximately 30% reduction in Piwi-bound cluster-mapping piRNAs (Supplementary Fig. 2h and Supplementary Data 4). However, a closer examination revealed that only 40% of the active clusters exhibited significant loss (more than 2.5 fold) of piRNAs mapping to them (Supplementary Fig. 2i and Supplementary Data 4). Only 50% of the clusters located on the chromosomal arms, excluding the ones on the heterochromatin and chromosome U, showed reduction in total piRNAs in HP1a-GLKD ovaries. Although the majority of piRNA clusters on chromosomal arms are spread in pericentric and telomeric regions, clusters showing piRNA loss were predominantly very close to telomeres and centromeres (Fig. 2c, Supplementary Fig. 2i). Among these, the clusters at chromosome X telomere (X:3833-27378) and 3R-tip (3R:27895169-27905030) exhibited the most severe total piRNA loss, while the clusters close to the centromeres of chromosomes X, 2, and 3 showed significant piRNA reduction, as well (Fig. 2c and Supplementary Fig. 2j). Additionally, the piRNAs mapping to the clusters at 42AB (2R:2144349-2386719) and 76D (3L:19841657-19861753) cytolocations were significantly decreased in HP1a-GLKD ovaries (Fig. 2c, Supplementary Fig. 2i). In contrast, HP1a-GLKD did not result in significant changes in the piRNA mapping to some major clusters in the pericentric and other regions (Fig. 2c).

The Piwi-bound cluster mapping piRNAs in HP1a-GLKD ovaries exhibited similar reduction trend as observed for total piRNAs. Only 30% clusters on chromosomal arms showed reduction in Piwi-bound piRNA mapping to them, in HP1a-GLKD ovaries (Supplementary Fig. 2h, and Supplementary Data 4). The clusters at chromosomes X and 3R tip showed highest reduction in Piwi-bound piRNAs, while the clusters close to centromeres on chromosome 2, pericentric cluster at 42AB, and cluster at 76D also showed significant reduction (Supplementary figure 2j). Some major clusters located in the pericentric

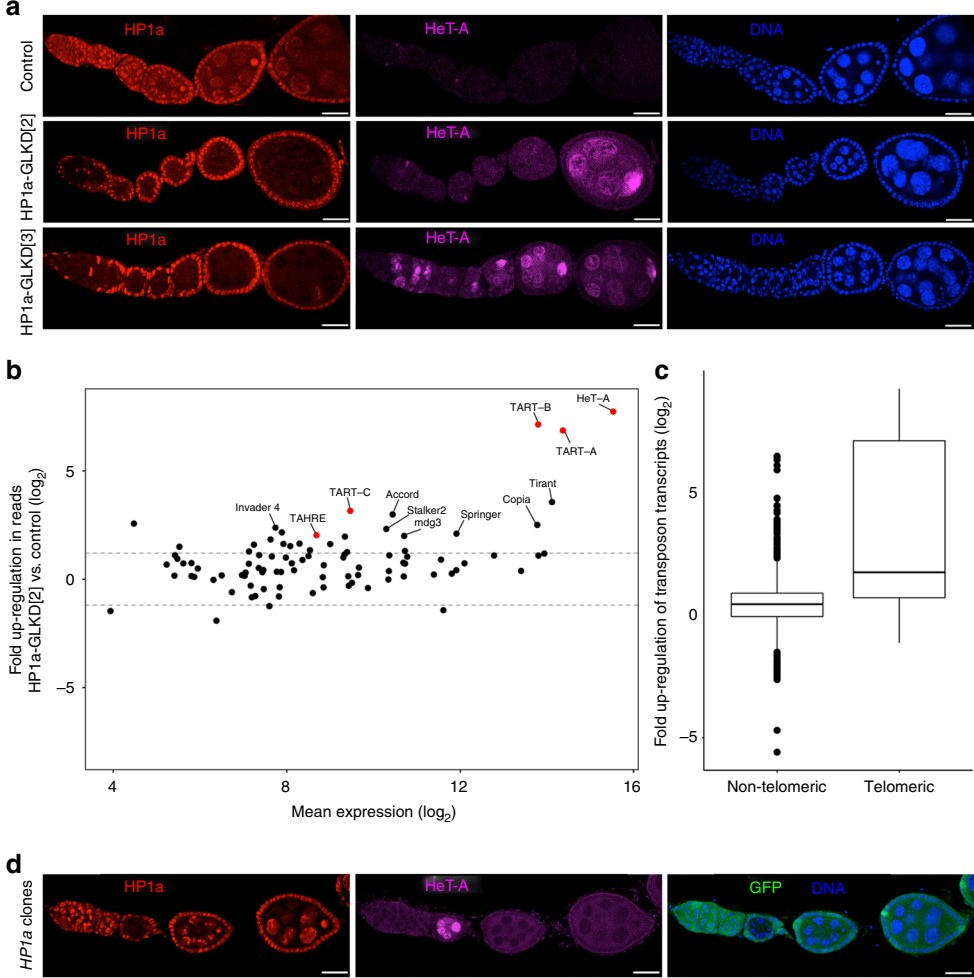

**Fig. 1** HP1a is required for the repression of selective transposons. **a** Representative images showing control and HP1a-GLKD fly ovaries stained for HP1a and HeT-A Gag proteins. Scale bars = 25 μm; n > 5. **b** Scatterplot showing the fold up-regulation of normalised RNA-seq reads mapping to canonical transposons in HP1a-GLKD compared to control ovaries. Telomeric transposons *HeT-A*, *TART*, and *TAHRE* are labelled in red. **c** Boxplot representing fold up-regulation in normalised RNA-seq reads mapping to telomeric and non-telomeric transposons in HP1a-GLKD vs. control. The middle line represents median. Box represents 25–75 percentile range called inter quartile range (IQR). Upper and lower whisker extend highest or lowest values till 1.5*IQR. Values above and below are outliers and plotted individually. **d** Ovaries with an HP1a mitotic clone (arrowhead) generated by FRT/FLP recombination stained for HP1a, GFP, and HeT-A Gag proteins. HP1a-null cells recapitulate show HeT-A upregulation. Scale bars = 25 μm; n = 3

regions were not affected (Supplementary figure 2h and 2i). These results suggest that HP1a is predominantly required for piRNA biogenesis from clusters close to telomeric and centromeric regions.

This prompted us to examine the genome-wide effects of HP1a-GLKD on the piRNAs uniquely mapping to the transposon insertions, including those in the piRNA clusters. The number of piRNAs mapping to transposon insertions outside the clusters were significantly less than those mapping to clusters (Supplementary Data 4). Compared with that in the control GFP-GLKD ovaries, the most severe piRNA loss was observed for the transposon insertions at cytolocations marking telomeric regions of chromosomes X, 2, and 3; cytolocations 1A, 60F, 61A, and 100E), and those at cytolocations marking centromeres of chromosomes X, 2, and 3 (i.e., cytolocations 20E, 20 F, 40, 80) (Fig. 2d, Supplementary Fig. 3c). While the HP1a-GLKD resulted in comparatively milder loss in the piRNAs mapping to the transposon insertions in several discrete cytolocations, such as 4, 18, 39F, 42AB, 73, and 76 (Fig. 2d). Some major piRNA producing loci located in the pericentric regions of chromosomes 2 and 3 (38, 39, 41) did not show any reduction in the piRNAs (Fig. 2d, Supplementary Fig. 3a, b and c). The Piwi-bound piRNAs uniquely mapping to transposon insertions, showed a similar pattern of piRNA loss in HP1a-GLKD ovaries

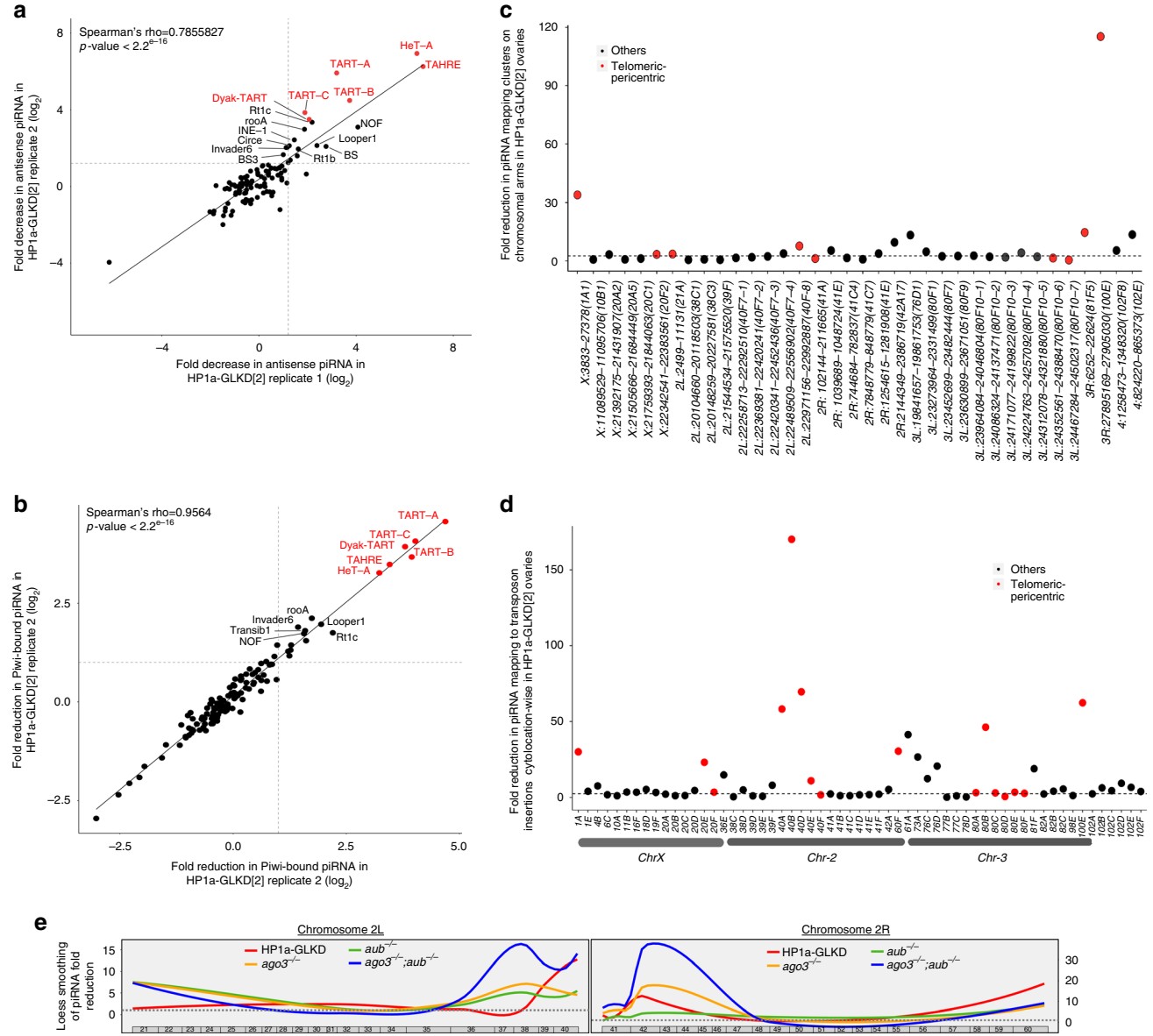

**Fig. 2** Loss of HP1a leads to the concomitant selective loss of piRNAs mapping to transposons. **a**, **b** Comparison of reduction in canonical transposon mapping piRNAs for **a** total cellular piRNAs in HP1a-GLKD[2] ovaries and **b** Piwi-bound piRNAs among two replicates of HP1a-GLKD[2] ovaries in comparison to control. Telomeric transposons, *HeT-A*, *TART*, and *TAHRE*, are labelled in red and correlation in piRNA loss among the replicates was estimated using Spearman's rank correlation test. **c** Fold reduction in the total cellular piRNAs mapping to clusters on the chromosomal arms. Clusters close to the telomeres and centromeres are represented in red. **d** Fold reduction in the piRNAs uniquely mapping to transposon insertions, for different cytolocations. The piRNAs uniquely mapping to transposon insertions were consolidated for each cytolocation and the cytolocations having more than 1000 uniquely-mapping piRNAs in control were analysed. **e** Loess smoothening of the reduction in the 23–29-nt reads uniquely mapping to the transposon insertions in chromosome arms 2L and 2R. Fold reductions in those reads in HP1a-GLKD (HP1a-GLKD[2] and HP1a-GLKD[3] average), *aub*, *ago3*, and *aub-ago3* ovaries compared with those in their respective controls are presented

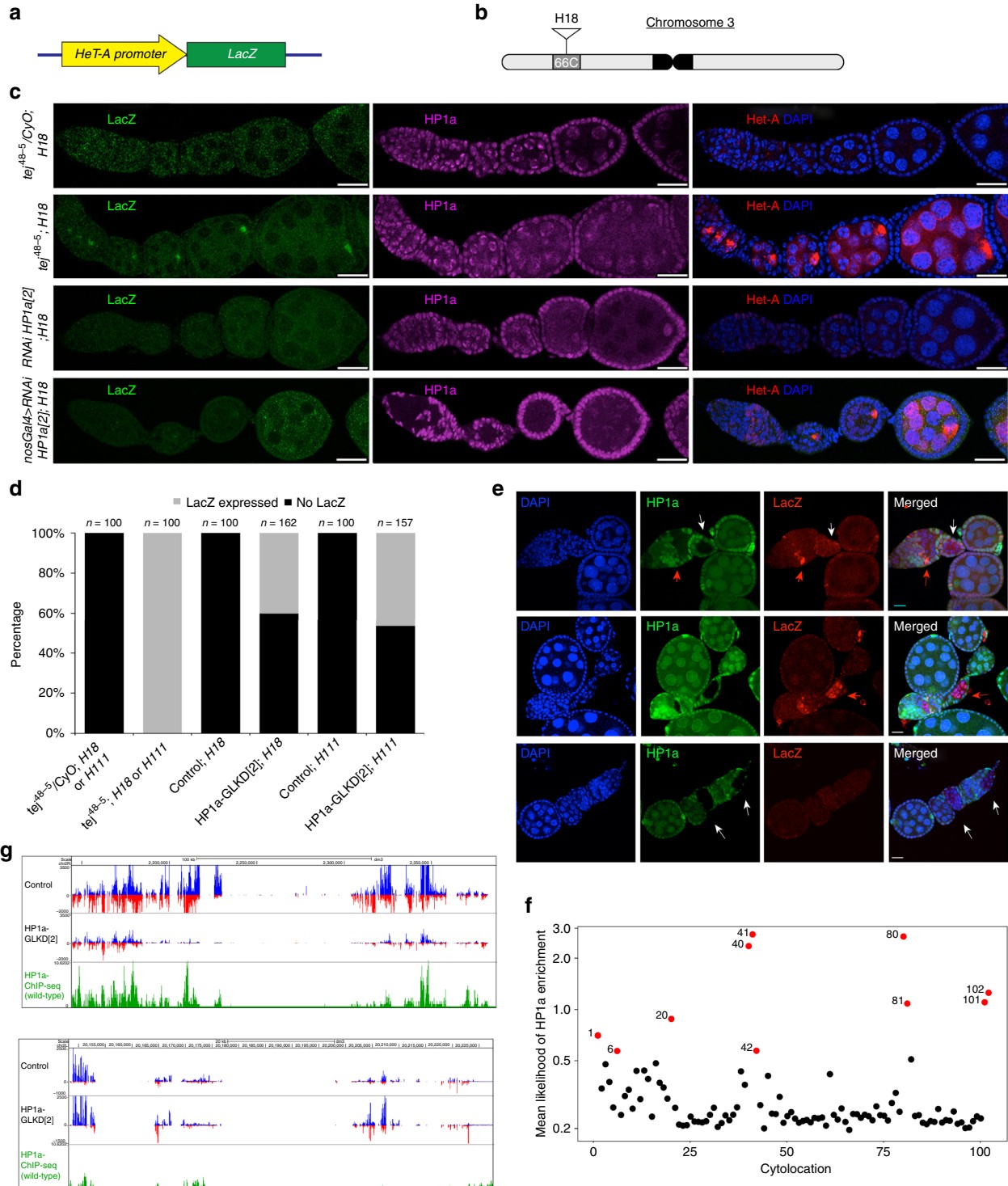

**Fig. 3** HP1a represses Het-A in a location-dependent manner **a** Schematic representation of the *HeT-A/lacZ* reporter construct. The *HeT-A* promoter is fused to the *lacZ* gene. **b** Schematic representation of the *HeT-A/lacZ* reporter insertion at the cytological position 66C (H18). **c** Representative confocal images of HP1a, LacZ, and HeT-A immunostaining in *tej*[48-5]/*CyO, tej*[48-5], and HP1a-GLKD ovaries. All genotypes harbour the *HeT-A/lazZ* transgene (H18 line). Three independent experiments were performed. Scale bars = 25 μm. **d** Bar graph showing the percentage of ovarioles expressing LacZ in *tej*[48-5]/ *CyO, tej*[48-5], and HP1a-GLKD genotypes bearing the *HeT-A/lacZ* transgene (H18 and H111). LacZ positive and negative ovarioles are represented by black and grey, respectively. *n* represents the number of ovarioles counted (*n* ≥ 100, from three independent experiments). **e** Representative confocal images of LacZ and HP1a immunostaining in the mitotic clones lacking HP1a. Red arrows and white arrows denote the HP1a clones with LacZ and those without LacZ expression, respectively. Scale bars = 15 μm. Three biological replicates were examined. **f** Scatterplot showing the mean likelihood of HP1a enrichment on chromosomal arms. Cytolocations marking telomeres and centromeres are represented in red. **g** UCSC browser screenshot showing the changes in total piRNAs and HP1a enrichment, following the normalisation with reads in the IgG ChIP-seq library, at cluster *42AB* (upper panel) and cluster *38C* (lower panel) in the control and HP1a-GLKD ovaries

(Supplementary Fig. 3d). These results suggest that HP1a is primarily required for the biogenesis of piRNAs from the telomeric, subtelomeric, and regions close to centromeres. Although, the full-length insertions of upregulated transposons upon HP1a-GLKD were spread out across genome, notably, 15 of

18 upregulated non-telomeric transposon families had majority of their insertion in pericentric regions. These included insetions from *invader4*, *springer*, *tirant*, *accord*, *copia*, *GATE*, and *gyspsy7* families (Supplementary Fig. 3d).

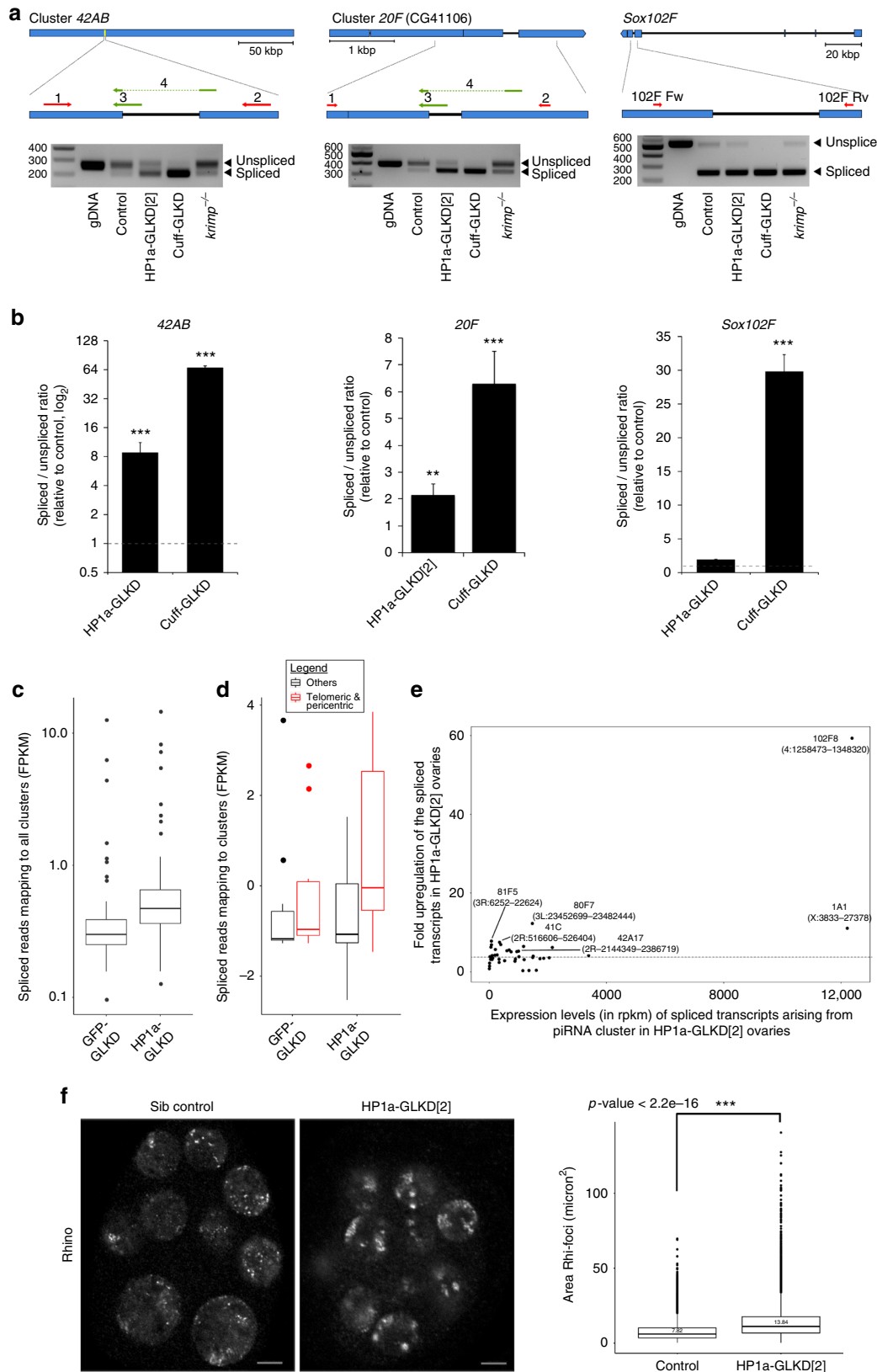

Furthermore, we compared the piRNA reduction pattern for annotated transposon insertions across genome in HP1a-GLKD ovaries with that in the mutants of the core piRNA components, *aub*, *ago3*, and their double mutant[9]. The piRNA loss in *aub*, *ago3*, and double mutant ovaries was more severe and widespread at the pericentric regions, compared to that in HP1a-GLKD ovaries (Fig. 2e). HP1a-GLKD did not result in net loss in total piRNA mapping to transposon insertions in cytolocations away from pericentric and telomeric regions, unlike other piRNA pathway mutants (Supplementary Fig. 3b). These results suggest that, unlike the core components of piRNA processing machinery, HP1a is predominantly required for piRNA biogenesis from regions close to telomeres and centromeres.

**HP1a represses transposon in location specific manner**. To investigate if HP1a function is required for piRNA-mediated repression of the transposons in location-dependent manner, we used a previously described transgenic reporter harbouring LacZ ORF under the *HeT-A* promoter, which was inserted away from the telomeric region (Fig. 3a, b)[30]. LacZ expression from this transgene was reported to be silenced via piRNA pathway in female germline cells[30]. In agreement with the previous report, we observed robust LacZ upregulation, as well as endogenous Het-A, in all egg chambers of a piRNA pathway mutant, *tej* (Fig. 3c, d)[24, 33]. Conversely, in HP1a-GLKD ovaries, only 40% of ovarioles expressed LacZ in at least one egg chamber, while almost all egg chambers exhibited endogenous HeT-A accumulation (Fig. 3c, d). Similarly, 45% of ovarioles containing an identical transgene inserted into another non-telomeric region, expressed LacZ in at least one egg chamber in the HP1a-GLKD ovaries (Fig. 3d and Supplementary Fig. 4a and 4b). To further confirm this observation, we performed clonal analysis of *HP1a* in the presence of *Het-A/LacZ* transgene, and observed that ~60% of the HP1a-lacking egg chambers expressed LacZ ($n = 20$), further supporting that HP1a functions more robustly for piRNA-mediated transposon repression at telomeric regions (Fig. 3e).

The effect of HP1a-GLKD on piRNAs suggested HP1a functions in piRNA pathway predominantly at regions close to telomeres and centromeres. This led us to examine genome-wide HP1a enrichment. We performed chromatin immunoprecipitation for HP1a using total ovarian lysates, which also contains somatic cells, followed by deep-sequencing (ChIP-seq). Consistent with previous reports, over 75% of HP1a peaks in wild type ovaries were observed in the transposon-enriched regions, while very few peaks were present in the genic regions (Supplementary Fig. 4c)[18,19]. HP1a occupancy was much higher at the

chromosome ends, and at the locations adjacent to the centromeres (cytolocations 1, 20, 40, 41, 42, 80, 81, 100, and 102; Fig. 3f and Supplementary Fig. 4d). HP1a-GLKD ovaries had maximum piRNA loss from the regions, which showed significant HP1a enrichment (Supplementary Fig. 4e). For example, HP1a-GLKD ovaries showed considerable reduction in the piRNAs mapping to the pericentric cluster at *42AB*, where HP1a was highly enriched, while piRNAs mapping to cluster at *38C* had no significant change, which had comparatively lower HP1a enrichment (Fig. 3g). This suggests that HP1a is required for piRNAs biogenesis from the regions it is enriched.

**HP1a-GLKD affects the splicing of the piRNA precursors**. HP1a is a nuclear protein, and the homologue of Rhino, which is required for proper transcription of piRNA precursors[6,9]. To understand how HP1a is required for piRNA biogenesis from clusters, we examined the effect of HP1a-GLKD on the steady-state levels of piRNA precursors using strand-specific qRT-PCR (Supplementary Fig. 5a). Among transcripts from three loci in the major cluster at *42AB*, expression level from one locus was significantly higher in HP1a-GLKD ovaries, while it remained unchanged for transcripts from two other loci (Supplementary Fig. 5a). We also observed an increase in the antisense transcript levels from the *HeT-A* loci, while the antisense *I-element* transcript levels remained unchanged (Supplementary Fig. 5a).

Recent studies addressing the role of Rhi, Cuff, and Del (RDC) complex showed that the suppression of piRNA precursor splicing by Rhi and Cuff is essential for proper piRNA biogenesis[6, 9]. To address any defects in piRNA precursor transcripts upon HP1a-GLKD, we examined the splicing of cluster transcripts originating from three piRNA clusters at, *42AB*, *20F*, and *sox102F*, by using a strand-specific RT-PCR[6, 9]. In agreement with the results from previous studies, the spliced transcripts from all these loci accumulated in the *cuff*-GLKD ovaries, but not in *krimp* mutant ovaries, which is a piRNA pathway component required for piRNA processing in female germline (Fig. 4a)[6,23]. HP1a-GLKD ovaries exhibited a clear accumulation of the spliced transcript from the *42AB* and *20F* clusters, which coincided with the remarkable reduction in piRNAs levels, mapping to these clusters (Fig. 4a). In contrast, the levels of spliced and unspliced forms from *sox102F*, which acts as a piRNA cluster in the germline, were comparable between controls and HP1a-GLKD ovaries (Fig. 4a)[9]. Consistently, the levels of piRNAs mapping to this cluster remained unchanged in HP1a-GLKD ovaries (Supplementary Fig. 2j and Supplementary Data 4). We quantified spliced and unspliced cluster transcripts using qRT-PCR, using primer sets to detect spliced or unspliced

**Fig. 4** HP1a is necessary for the proper splicing of piRNA precursors from selected clusters. **a** Amplification of regions from piRNA clusters at *42AB*, *20F*, and *sox102F*. Representative gel images of RT-PCRs show amplification and levels of the spliced and unspliced transcripts originating from clusters *42AB*, *20F*, and *sox102F* in HP1a-GLKD[2] ovaries. *cuff*-GLKD and *krimp* mutants were used as positive and negative controls, respectively. Three independent experiments were performed. **b** Spliced to unspliced transcript level ratio (SSR) calculated for the piRNA clusters *42AB*, *20F*, and *sox102F*. qRT-PCR was conducted with a primer set detecting either unspliced or spliced transcript. Three independent experiments were performed. The significance of difference in levels of spliced transcript between control and HP1a-GLKD[2] and between control and cuff-GLKD, ovaries was statistically tested using Mann-Whitney *U* test. $n = 3$, three independent biological replicates, error bars represent standard error for all replicates. *p* values: ** represent *p* value <0.05 and *** represent *p* value <0.005. **c** Boxplot showing the changes in the average FPKM values of spliced transcripts mapping to the piRNA clusters in the HP1a-GLKD[2] ovaries vs. the controls. **d** The average FPKM values of spliced transcripts mapping to the piRNA clusters in cytolocations 1, 20, 21, 40, 41, 60, 61, 80, 810, 100, and 102 with that in the clusters in rest of genome, in control and HP1a-GLKD[2] ovaries. **e** Fold increase in the spliced transcript coming from major piRNA clusters in HP1a-GLKD ovaries. **f** Representative confocal images of Rhi immunostaining in the control and HP1a-GLKD ovaries (left panel) and comparison of Rhi foci sizes between the control and HP1a-GLKD[2] ovaries (right panel). Scale bar = 5 µm. Sixty egg-chambers from three biological replicates were analysed for the Rhi foci sizes. Statistical significance in differences of Rhi foci between the genotypes was examined using Kolmogorov-Smirnov test. For the boxplots in **c**–**e**, the middle line represents median. Box represents 25–75 percentile range called inter quartile range (IQR). Upper and lower whisker extend highest or lowest values till 1.5 * IQR. Values above and below are outliers and plotted individually

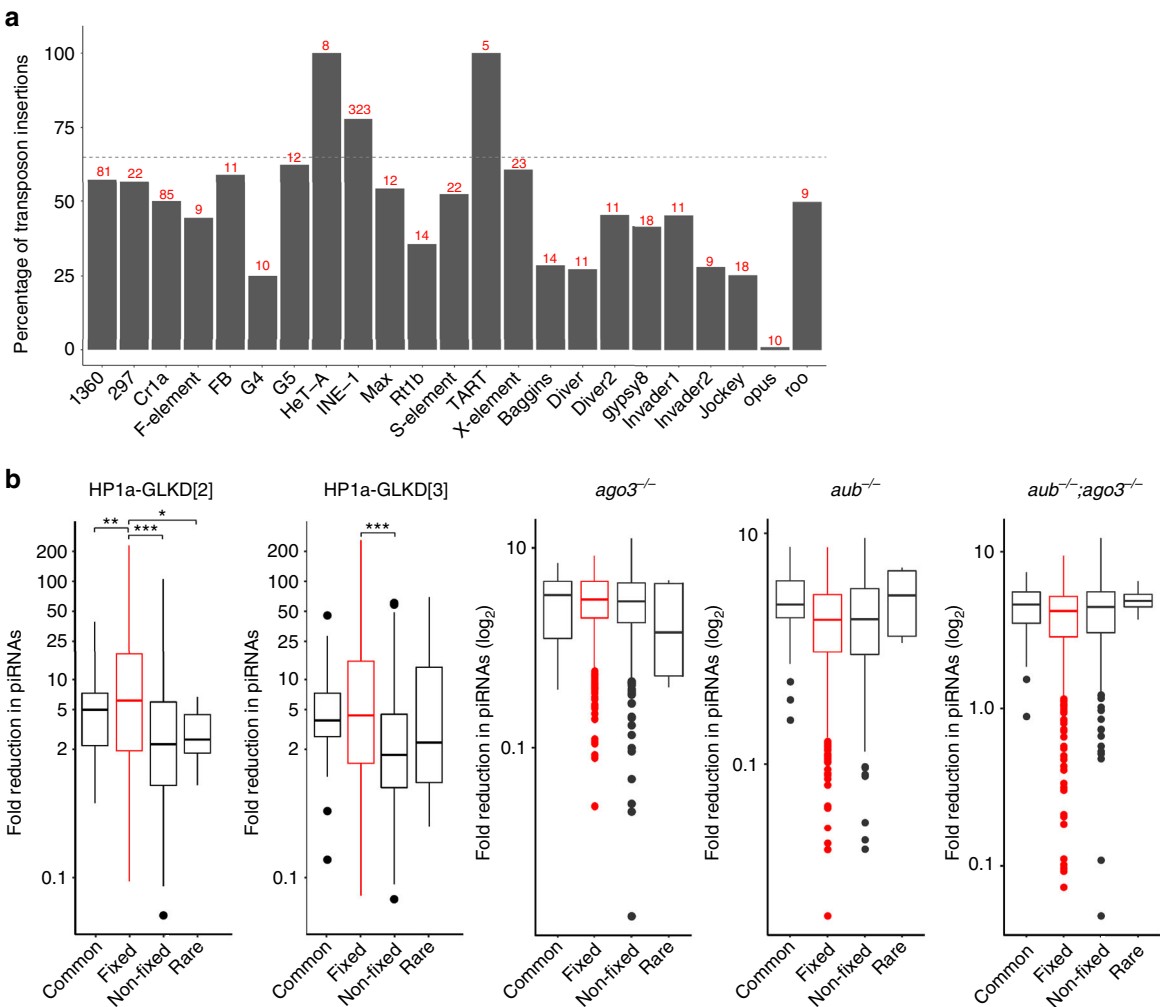

**Fig. 5** HP1a is required for the biogenesis of piRNAs mapping to evolutionarily older transposons. **a** The percentage of the transposons insertions from major transposon families that showed at least three-fold loss in the uniquely-mapping piRNAs in the HP1a-GLKD ovaries. The average reduction between the 23-29-nt read levels in the HP1a-GLKD[2] and HP1a-GLKD[3] ovaries was plotted, and the insertions targeted by at least 20 unique piRNAs were considered. The number in red represents number of such insertions for each transposon family. **b** Boxplots showing the reduction in the piRNAs mapping to the evolutionarily fixed, common, and rare transposon insertions, based on their occurrence in population, in the HP1a-GLKD and *ago3*, *aub*, and *aub-ago3* double mutants. The significance of higher reduction of piRNAs mapping to evolutionarily fixed transposon insertions in comparison with those mapping to common or rare insertions was examined using Mann-Whitney *U*-test. The middle line represents median. Box represents 25–75 percentile range called inter quartile range (IQR). Upper and lower whisker extend highest or lowest values till 1.5 * IQR. Values above and below are outliers and plotted individually

forms, separately (Fig. 4b). The ratio of spliced to unspliced transcripts levels increased for the cluster *42AB* and *20F* transcripts, while this ratio for the *sox102F* transcripts did not change (Fig. 4b). Furthermore, we examined any increase in spliced forms from other clusters, using our RNA-seq data. In the HP1a-GLKD ovaries, a significant increase in reads mapping to the spliced piRNA cluster transcripts arising from many clusters in the telomeric and centromeric regions was observed (Fig. 4c, d). We observed maximum increase in the levels of spliced cluster transcripts arising from *HeT-A* cluster at chr 4 (102F8, 4:1258473-1384320) and chr X telomere (1A, X:3833-27378) and clusters at centromeric region of chr 2 and 3, these clusters also showed severe reduction in piRNAs in HP1a-GLKD ovaries (Figs. 2c and 4e). Conversely, we did not observe any significant changes in the spliced transcript levels of the protein-coding genes in the HP1a-GLKD ovaries (Supplementary Fig. 5b).

Rhi, a HP1a homologue, together with its interacting partner cuff, suppresses the piRNA precursor splicing. A decrease in the Rhi accumulation at the cluster loci correlates with an increase in the splicing rate of piRNAs precursors[6,9,34]. To study whether HP1a functions to repress the splicing through Rhi, we examined effect of HP1a-GLKD on Rhi nuclear localisation by immunostaining, and its occupancy at piRNA clusters using ChIP. As previously reported, Rhi is localised as discrete foci across the nucleus in the controls[7,8]. In the HP1a-GLKD ovaries, however, Rhi foci appeared more aggregated, and were closer to nuclear periphery (Fig. 4f). Additionally, in the HP1a-GLKD ovaries, Rhi occupancy was reduced at the clusters at X chromosome (X:3833-27378) and *42AB* (2R:2144349-2450369), but remained unchanged at *38C*, which did not show any reduction in piRNAs (Supplementary Fig. 5d, 2c and Supplementary Data 4), suggesting HP1a likely stabilises Rhi for proper repression of the splicing

from the piRNA clusters, for which we observed piRNA loss in HP1a-GLKD ovaries.

HP1a-GLKD did not result in any significant changes in the localisation of piRNA pathway components Aub, Ago3, and Tej at nuage, the piRNA processing site (Supplementary Fig. 5c)[23], indicating that the assembly of the piRNA pathway components at nuage does not depend on HP1a. Nuclear Piwi localisation was also not affected in the HP1a-GLKD ovaries. In addition, steady state transcript expression levels of the piRNA pathway genes, were comparable between control and HP1a-GLKD ovaries (Supplementary Fig. 5c and e).

Taken together, increase in splicing of cluster transcripts and piRNA loss upon HP1a-GLKD from the clusters enriched for HP1a suggests that HP1a functions upstream to piRNA processing, by repressing splicing of transcripts from the regions it is enriched.

**HP1a maintains targeting of evolutionarily old transposons**. Previous studies showed that evolutionary fixed transposon insertions, i.e., those present in the majority of individuals within a population, are enriched in the pericentric regions[35]. This prompted us to examine the effect of HP1a-GLKD on the piRNAs uniquely-mapping to transposon insertions, which belong to different transposon families (Fig. 5a, Supplementary Fig. 6a, and Supplementary Data 5). Remarkably, HP1a-GLKD ovaries showed a significant loss in total and Piwi-bound piRNAs mapping to approximately 83% of the *INE-1* insertions. Almost all *INE-1* insertions are considered evolutionarily fixed, and majority of these are present in the pericentric region[36,37] (Fig. 5a and Supplementary Fig. 6a). We compared loss in the piRNA mapping to all transposon insertions categorised as fixed with those mapping to common and rare insertions (Fig. 5b)[35,38]. Expectedly, we observed more severe loss in piRNA mapping to fixed transposon insertions compared to those mapping to common and rare ones, upon HP1a-GLKD (Fig. 5b and Supplementary Fig. 6b). In *ago3, aub*, and double mutants, the loss of total piRNAs mapping to the fixed and other transposon insertions was comparable (Fig. 5b).

A previous study, using a different approach, suggested evolutionarily old transposons show increase in sequence divergence in the piRNAs mapping to them, and suggested that Piwi-bound piRNAs are required to repress evolutionarily old transposons[29]. More than half of these evolutionarily old transposons, including *HeT-A, TART, TAHRE, invader4, GATE, mdg3*, and *copia*, exhibited a significant reduction in the total and Piwi-bound piRNAs in the HP1a-GLKD ovaries (Fig. 2a, b, and Supplementary Fig. 2f)[29]. We also investigated the correlation between the sequence divergence in the piRNAs mapping to the canonical transposon and severity of piRNA loss mapping to each transposon family upon HP1a-GLKD (Supplementary Fig. 6c). Transposons targeted by more number of piRNAs with 2–3 nucleotide mismatch exhibited more severe piRNA reduction, and comparatively greater upregulation, suggesting HP1a-GLKD predominantly leads to loss in piRNA targeting evolutionarily-older transposons.

## Discussion

HP1a is involved in many cellular processes ranging from the maintenance of heterochromatin and telomeres to the regulation of gene expression in somatic cells[39]. The role of HP1a in transposon repression has been described in the germline and somatic cells of the female *Drosophila* gonads[10,13,16]. In the female germline, HP1a was shown to repress transposon expression and is believed to exert its functions downstream of Piwi-piRISC, although its function in the piRNA pathway in

germline has not been investigated in detail[13]. In this study, we employ RNAi-based approach to knockdown HP1a in female germline cells. Although we cannot exclude a possibility of milder effect due to incomplete knockdown, we nevertheless show that HP1a also functions in piRNA biogenesis and upstream to piRNA-mediated transcriptional repression.

We show that HP1a-GLKD resulted in piRNA loss arising from a limited number of clusters, and in loss of piRNAs targeting a subset of full length transposons in adult female germ-lime cells (Fig. 2). Ronsseray and colleagues reported that HP1a-GLKD in *Drosophila* embryos causes piRNAs loss against telomeric transposons and cluster at 3R-tip, suggesting restricted HP1a function in the piRNA pathway in embryo as well[40]. The same cluster at 3R-tip, designated as cluster at *100E*, consistently shows most severe piRNA loss among all clusters upon HP1a-GLKD in the adult germline (Fig. 2c). However, they did not observe reduction in piRNAs mapping to cluster at *42AB* or other regions. In the embryonic stages, the epigenetic state of the clusters and ping-pong amplification are established, while the adult gonads have a completely functional piRNA pathway[24,33,41]. This might lead to differences in observation about HP1a function in piRNA pathway between different development stages. In addition, an earlier study by Ronsseray and colleagues also showed reduction in the piRNAs from a transgene inserted in X-TAS in HP1a heterozygous ovaries, suggesting HP1a role in piRNA biogenesis[42].

Our study suggested more robust germline function of HP1a in piRNA pathway at regions close to telomeres and centromeres. This is likely explained by high HP1a enrichment in these regions. However, we cannot rule out somatic contribution to HP1a enrichment as the ChIP-seq experiment was carried out using total ovarian lysates including somatic cells. Similarly, *HP1a* knockdown in ovarian somatic cells also resulted in the derepression of a limited number of transposons[30]. The HP1a enrichment at the telomeric and pericentric regions was also observed in *Drosophila* somatic cells and embryos, suggesting that HP1a maintains telomeric and pericentric regions in different *Drosophila* cell types[43–45]. However, we did not detect an increase in the HP1a enrichment at the ends of chromosomes 2 and 3, possibly due to the lack of the relevant telomeric sequence information.

Genetic analysis using *HeT-A/lacZ* reporter further demonstrated that HP1a-role in downstream piRNA-mediated transposon repression is more robust at the telomeric regions whereas lacZ derepression is observed only in a limited number of HP1a-GLKD ovarioles.

HP1a-GLKD led to the upregulation of cluster transcript splicing, perturbation of the Rhi localisation, and the reduction of the Rhi occupancy at clusters in the telomeric and regions close to centromeres (Fig. 4f, Supplementary Fig. 5c, d). These results suggest that HP1a functions upstream to piRNA processing, for the production of proper piRNA precursors from these regions by stabilising Rhi (Supplementary Fig. 5c). Notably, however, two other regions of examined *42AB* cluster did not show such increase of splicing (Supplementary Fig. 5a). This kind of heterogeneity was also described for cluster transcript arising from *42AB* in *cuff* mutants[6,46]. GLKD of a HP1a partner, Piwi, in embryonic stages is reported to cause accumulation of the spliced cluster transcripts from bi-directional piRNA clusters[31]. HP1a may interact with other proteins, such as Rhi and Piwi, to repress the splicing of piRNA precursors at the heterochromatic regions.

Our results also suggest that HP1a functions downstream to piRNA processing for repression of several transposons, such as *tirant, accord*, and *transib1*. The *tirant* insertions are located primarily away from the piRNA clusters and are found between genes or on the opposite strands to genes, and all *tirant* copies

have been classified as evolutionarily fixed[35]. This strengthens the importance of HP1a during the repression of the evolutionarily old transposons, which are enriched in the pericentric regions with the HP1a enrichment[35]. These evolutionary old or fixed transposon insertions were proposed to be silenced in the germline at the transcriptional level with fewer piRNAs (Supplementary Data 5)[29,47]. The average number of uniquely mapping piRNAs targeting *INE-1* was shown to be lower than that of piRNAs targeting other transposon families (Supplementary Data 5)[48].

The HP1a importance in repression evolutionarily old transposons and their enrichment at perincetric regions might contribute to the compartmentalised function of HP1a in piRNA biogeneis. The highly conserved HP1a may have adapted to silence the evolutionary older transposons in the pericentric and telomeric regions. In contrast, the functions of newly-evolved and *Drosophila*-specific proteins such as Rhi and Cuff in the maintenance of the cluster transcripts genome-wide may indicate the adaptation of the piRNA pathway to novel transposon threats and reproductive isolation[48]. An extensive analysis of HP1a interacting partners and epigenetic markers throughout the genome, especially in the pericentric and telomeric regions, may reveal unique piRNA production mechanisms in different chromosomal compartments.

## Methods

**Fly strains**. *Drosophila melanogaster* strains and alleles used in this study were *su (var)2–5[05]*, RNAiHP1a[2] (TRiP BL#36792), RNAiHP1a[3] (TRiP BL#33400), RNAiGFP (TRiP BL#41556), RNAicuff (BL#35182), *krimp[f06583]* [23] and *tej[48-5]* [24]. The following drivers were used for RNAi induction in the germline: *NGT40* and/or *nosGal4-VP16*, and *trafficjam*-Gal4. *HeT-A/lacZ* reporter flies H18 and H111 were obtained from Dr. Alla Kalmykova[30], Institute of Molecular Genetics, Moscow, Myc-Piwi transgenic flies were obtained from Dr. Haifan Lin[49], Yale University, Yale. The flies were grown on standard cornmeal-agar medium at 25 °C.

**FRT recombination**. Mitotic germline clones of the loss-of-function mutants were generated using the Flippase/FRT recombination target (FLP-FRT) technique[26]. For clone generation, *su(var)205[05]* was recombined with the *FRT40A* allele and crossed with *hsFLP; Ubi-GFP FRT40A* flies. Pupae were heat shocked at 37 °C for 1 h, twice daily for two consecutive days, and the adults were dissected 6 days after the heat-shock.

**Immunostaining and antibodies**. Ovaries were fixed in the 4% paraformaldehyde (PFA) for 5 min, after dissection. Subsequently, the ovaries were washed in PBS for 30 min followed by blocking in 5% goat seurm for 1 h. After blocking, the ovaries were incubated in primary antibodies for 4 h at room temperature, followed by washing in PBS-triton X-100 (0.2%) for 1 h, with several changes of wash solution. The ovaries were then incubated with secondary antibodies for 4 h at room temperature followed by washing as described above. The ovaries were mounted in Vectashield (Vector Labs), for examination under a fluorescence microscope.

Antibodies used for immunostaining were anti-HP1a (mouse, 1:50; Developmental Studies Hybridoma Bank), anti-HP1a (rabbit, 1:200; Covance PRB293C), anti-HeT-A gag (guinea pig, 1:1000; a gift from Dr. Yikang Rong), anti-I-element gag (rabbit, 1:1000; a gift from Dr. David Finnegan), anti-Aub, (mouse, 1:1000; a gift from Dr. Siomi), anti-Ago3 (rabbit, 1:500; a gift from Dr. Dahua Chen), anti-Rhino (1:1000; a gift from Dr. William Theurkauf), anti-Tej (rabbit, 1:2500; in-house), anti-Piwi (mouse, 1:1; a gift from Dr. Siomi), anti-LacZ, (rabbit, 1:10,000; a gift from Dr. Yu Cai), anti-GFP (chicken, 1:1000; Abcam ab13970), and anti-Myc (mouse, 1:1000; Wako 017-21876). Alexa Fluor- (488, 555, or 633) conjugated goat anti-mouse, anti-rabbit, anti-guinea pig, and anti-chicken immunoglobulin (1:200; Invitrogen) were used as secondary antibodies. Images were captured with a Zeiss LSM Excitor upright or Leica SPE-II upright confocal microscope and processed using ImageJ and Adobe Photoshop CS6.

**RNA extraction, reverse transcription, and qPCR**. Ovaries were dissected in cold PBS and total RNA was extracted using TRIzol (Invitrogen). Before reverse transcription, the RNA was treated with Turbo DNase (Ambion) following the manufacturer's instructions. Reverse transcription was carried out with oligo(dT)$_{20}$ or strand specific primers (Supplementary Data 6) using the Superscript III Reverse Transcriptase system (Invitrogen), following manufacturer's instructions. Quantitative-PCRs (qPCRs) were carried out using the KAPA SYBR FAST qPCR kit (KAPA Biosystem) on Applied Biosystem 7900HT Fast Real-Time PCR system. Transcript levels were quantified using the $\Delta\Delta C_T$ method and normalised to *rp49*.

All experiments were carried out at least in triplicate, with the average and standard error shown. Information about the primers is provided in Supplementary Data 6.

**Transcriptome analysis**. Total RNA from the control and HP1a-GLKD ovaries was isolated using TRIzol (Invitrogen). Total RNA libraries were prepared using the Tru-Seq RNA Library Preparation Version 2 kit, with standard Illumina adaptors. They were subjected to pairing end sequencing, with 101-nt read length. Adaptor and low-quality reads were removed using Trimmomatic[50]. The reads were mapped to the *D. melanogaster* genome (release 5) using Tophat2[51]. Differential expression of the genes and transposon insertions was assessed using geometric normalisation. The libraries were aligned to the canonical transposon sequence (Flybase). The number of reads mapping to each transposon was calculated and the expression levels were normalised using the same parameters used for the library mapping to the genome. To analyse the splicing in cluster transcripts, the Tophat2 mapped bam files were used for the transcript prediction using Trinity[52]. The transcripts were mapped back to *D. melanogaster* genome (release 5), with blast. Splicing in the cluster mapping transcripts was estimated after the alignment. The quantitation of transcripts was performed using rsem[53].

**Chromatin immunoprecipitation (ChIP) analysis**. For each ChIP, ~200 pairs of ovaries were dissected and fixed with 1.8% paraformaldehyde for 10 min at room temperature. Fixed ovaries were resuspended in cell lysis buffer [5 mM PIPES pH 8.0, 85 mM KCl, 0.5% NP-40, and EDTA-free complete protease inhibitor (Roche) and disrupted using a dounce homogeniser. Isolated nuclei were washed and resuspended in SDS lysis buffer (ChIP Assay kit, Millipore). The nuclear lysate was fragmented using a water bath sonicator (BioRuptor, Diagenode) to fragment sizes of 200–800 bp. The nuclear lysate was pre-cleared with protein A agarose beads for 30 min followed by IP with an anti-HP1a antibody (rabbit, Covance) overnight at 4 °C. Washes were performed according to manufacturer instruction and immunocomplexes were eluted from the beads and de-crosslinked. Immunoprecipitated DNA was recovered by phenol/chloroform extraction and ethanol precipitation. DNA was resuspended in 30 μl water and used for qPCR reactions. We performed single end sequencing of HP1a ChIP derived fragments.

**RNA immunoprecipitation (RIP)**. For each RIP, ~250 pairs of ovaries were dissected and lysed in 250 μl lysis buffer (0.5 M HEPES KOH, pH 7.3, 150 mM NaCl, 5 mM MgCl$_2$, 10% glycerol, 1% Triton X-100, complete protease inhibitors (Roche), 100 U RNaseOUT (Invitrogen), 1 mM DTT, 1 mM EDTA, and 0.1 mM PMSF) and spun for 2 min at maximum speed in a table top centrifuge at 4 °C. The resultant supernatant was stored on ice. The pellet was re-extracted three times with 250 μl of lysis buffer and the resultant nuclei were pooled together.

For Piwi IP, two sample replicates were prepared for Myc-Piwi RIP in wild-type and HP1a-GLKD ovaries and one for IgG IP. A monoclonal mouse antibody against Myc (Wako 017-21876) was used. After the IP procedure, RNA was extracted from the magnetic beads using TRIzol. Then, 100 ng Piwi IP RNA was radioactively labelled using a CIP-kinase assay with γP$^{32}$-ATP and small RNAs were cloned as outlined below.

**Small RNA library preparation**. Ovarian RNA, extracted using TRIzol, was depleted for 2S rRNA using an anti-2S rRNA oligo, described in Li et al.[54,55]. rRNA-depleted and IP samples were subjected to electrophoresis, using a denaturing polyacrylamide gel and small RNAs ranging from 18–29 nt were excised and eluted. The eluted RNAs subsequently were subjected to ligation of 3′- and 5′-barcoded adaptors containing 4 random nucleotides at the ends to reduce ligation biases followed by reverse transcription, PCR amplification and sequencing in on a Illumina HiSeq 4000 instrument.

**Statistical analysis**. For all statistical comparisons, we first analysed the data distribution with the Shapiro–Wilk test for normality. We used appropriate statistical tests, based on the comparison and population. For example, two unpaired samples with non-normal distribution were compared using two-tailed KS-test (Fig. 4f and 5b). All other analysis including Pearson correlation, loess smoothing and others were performed using R[56].

**Small RNA-seq and ChIP-seq data analysis**. Adaptors and low quality reads were removed prior to alignment. The small RNAs were mapped to canonical transposon sequences, permitting two mismatches. We only considered 23–29 nt reads as piRNAs. The reads were mapped to the *D. melanogaster* genome (release 5) without any mismatches. We only considered reads that were uniquely mapping to the clusters and transposon insertions. The coordinated for transposon insertions was obtained from Flybase. The small RNA libraries made from the control and HP1a-GLKD ovaries were normalised by depth and by per million miRNA mapping reads. Both normalisation methods showed comparable results; we subsequently used miRNA normalisation for all analyses in this study. For the ping-pong analysis, ping-pong z-scores were calculated by the probability of a 10-nt overlap minus the average of probability of other overlaps (2–9, 11–20-nt overlap) divided by its standard deviation of the probability of other overlaps. The reads were mapped to 10-kb windows on each chromosome arm using BEDtools[57]. The

sequencing depth for second set of replicates (replicate 2; r2) for each GLKD from RNAiGFP, RNAiHP1a[2], and RNAiHP1a[3], was higher than (replicate 1; r1). Hence, replicate set 1 was used to show congruence in piRNA loss pattern in Fig. 2. Subsequently, data from replicate set 2 was used for small RNAs analyses in the rest of the results.

ChIP-Seq analysis, adaptors, and bad quality reads were removed prior to sequencing. The reads were analysed against the *D. melanogaster* genome (release 5) using Bowtie[58]. The peaks were predicted using MACS2[59]. To analyse the HP1a mapping reads on the genome, IgG mapping reads were subtracted from HP1a ChIP-Seq reads after normalisation of both libraries by depth.

**Data availability**. Deep sequencing data generated in this study is deposited at NCBI SRA, with the accession number SRP095292[32]. The details of these and other libraries used in the study are listed in Supplementary Data 1.

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

## Acknowledgements

We are grateful to Dr. Alla Kalmykova, Institute of Molecular Genetics, the Russian Academy of Sciences, Moscow for providing *HeT-A/lacZ* transgenic flies, and to Dr Yikang Rong, the Sun Yat-Sen University, Guangzho and Dr. David Finnegan, the University of Edinburgh, Edinburgh for providing HeT-A Gag and I-element ORF antibodies, respectively. We would like to thank the Bloomington Drosophila Stock Center and Transgenic RNAi Project for providing fly stocks. We thank Dr. Nicholas Tolwinski for critical reading and suggestions. This work was supported by the Temasek Life Sciences Laboratory and the Singapore Millennium Foundation, the Naito Foundation, and the Japan Society for the Pormotion of Science (JSPS) Grants-in-aid for Scientific Research (KAKENHI) 26291048 (T.K.); NRF Fellowship NRF2011NRF-NRFF001-042 from the National Research Foundation Singapore (K.O.).

## Author contributions

A.A., R.Y.W.T., T.K., and K.O. designed the experiments. R.Y.W.T. and A.A. performed experiments and analysed data. A.A. and V.S. performed bioinformatics analysis. R.Y.W.T., A.A., and T.K. wrote manuscript.

## Additional information

**Competing interests:** The authors declare no competing interests.

