## [Peer Review File · Nature Communications]

Reviewers' comments:

Reviewer #1 (Remarks to the Author):

This work uses a combination of genetic experiments and genomics with bioinformatics to understand the role of HP1a in transposon silencing and piRNA production in *Drosophila*. The authors find that HP1a represses a subset of transposons, predominantly located in telomeric and pericentric regions. This repression is probably due to HP1a-dependent production of piRNAs corresponding to these transposons. This paper suggests that HP1a represses splicing of piRNA precursors derived from telomeric and pericentric regions, thereby ensuring the production of piRNAs corresponding to transposons located in telomeric and pericentric regions. Overall, the paper has relevant information that will be useful for people working on piRNA biogenesis and piRNA-mediated transposon silencing in *Drosophila*. However, comments below indicate aspects that need significant revision before publication. This revision is important to determine the actual impact of the work.

Major comments:

1. This paper lacks experimental data as to how HP1a robustly represses transposons located in telomeric and pericentric regions, whereas not effectively at non-telomeric and non-pericentric regions. Where does the specificity come from?
2. This paper lacks experimental data as to how HP1a represses splicing of piRNA precursors and how HP1a recognizes and regulates splicing of piRNA precursors derived from telomeric and pericentric regions. Where does the specificity come from?
3. Figure 4: Sox102F transcripts are almost completely spliced in control. Thus, it is not a good control to show that HP1a KD does not result in increase in splicing of the transcripts. The authors shall consider analyzing other cluster transcripts.
4. It is hard to understand how the data shown in Figure 4b are computed based on the results shown in Figure 4a. Particularly, for 20F. Is the difference (with such large error bars) really significant?
5. It is difficult to follow Figure 2c and d, Figure 4e, Sup Figure 2h. It is difficult to understand which clusters are located in telomeric and pericentric regions.
6. The paper seems to have been written in a hurry and some aspects of it need to be improved. The paper needs revising in English grammar and syntax

Reviewer #2 (Remarks to the Author):

The study by Teo et al is of interest for different fields including piRNA biology, chromatin structure and gene regulation. This study addresses an important question in the piRNA field, namely the role of HP1 in the piRNA production in the germline. Using whole genome data the authors find that HP1 knockdown in *Drosophila* germline caused substantial decrease in piRNA production by telomeric and pericentric piRNA clusters accompanied by accumulation of spliced piRNA precursors from these loci. Given that HP1 binds not only piRNA targets, but also piRNA cluster regions, it is surprising that the role of HP1 in the piRNA production has not been explored yet. Moreover, this study adds a novel information about an additional role of highly conserved HP1 chromatin protein in the piRNA biogenesis. While the manuscript does not provide mechanistic insights into the particular role of HP1 in the piRNA biogenesis, I find it overall of broad interest. Despite the fact,

that the work is well supported by the experimental data, I would suggest a serious revision of manuscript to better present these data, to avoid wrong conclusions and to make the story more convincing.

In Abstract and along all the manuscript, the authors write "loss of HP1" implying not null mutation (which is lethal) but HP1 RNAi knockdown. However, efficiency of different siRNA constructs in knockdown experiments varies highly and is not 100%. Moreover, nanos promoter driving GAL4 expression has specific expression pattern during oogenesis. Thus, these data describe knockdown the efficiency of which is very difficult to estimate because ovaries are a complex tissue containing both somatic and germ cells. It should be taken into consideration and rephrased.

Results, First section contains observations, which mainly confirm previous results. In Wang and Elgin 2011 strong upregulation of not only HeT-A, but also Blood, Burdock and Bari was observed upon HP1 GLKD (strains from VDRC were used), however, in Tao et al derepression of these TEs was not detected (Fig1b, lanes 71-75). There is also inconsistency in TE derepression data between Fig1b (RNA-seq) and Supplementary Fig1c (RT-PCR). Using clonal analysis, the authors confirmed the activation of HeT-A but did not analyze other TEs from Wang and Elgin. I-element is used as a TE which is not activated in HP1_KD. However, it is not the best choice, because I-element is not affected by piwi mutations as well (Klenov et al 2011). Thus, the conclusion "that HP1a is required in the germline cells to repress a subset of the transposons" is not quite correct. I suppose that efficiency of the HP1_KD was not very high (see below). The authors have to discuss all these discrepancies and soften the conclusion. Actually, only telomeric HeTA seems to be upregulated. This is not surprising because HeT-A is the most sensitive among all TEs to the piRNA pathway disruption. The authors should confirm the conclusion about telomeric TEs upregulation by normalization to the genomic copy numbers. Even in the strains with similar genetic background the copy number of telomeric TEs can be different. Thus, PCR on genomic DNAs of all genotypes used in the study with primers specific to telomeric TEs is essential for normalization of RNA-seq and smallRNAseq data.

Lines 83-87. Previously, TEs were subdivided into 3 types, germline, somatic and intermediate relative to their ability to be overexpressed upon piwi depletion (Malone et al). The data described here most likely provide confirmation to these results. In Supp Fig1f, expression levels of Idefix and gypsy should be included as a control to the efficiency of HP1_KD in follicular cells. These TEs should be overexpressed because Piwi-mediated transcriptional silencing requires the HP1 (Le Thomas et al., 2013; Rozhkov et al., 2013).

Fig2b – Both axes are designated as replicate 2, needs correction.

I didn't find the numbers of Supplementary Tables neither in file names nor in the Tables themselves.

Lines 119-135; description of cluster-derived piRNAs is very confusing mostly due to the absence of their accurate designations and poor pictures. FigS2h and Table S4 show genomic coordinates of the piRNA clusters, while in the text, the authors refer to telomeric, pericentric, 38C, 10A, 42AB and 98F clusters. The readers will not be able to read all these data. The authors should use similar designations for all pictures or at least add the designations of the clusters mentioned in the text to Suppl. Fig2h. For example, in Brennecke et al 2007 the clusters were termed as ML# (master locus #). The phrase "In addition to telomeric clusters, piRNAs mapping to clusters near centromeric regions also exhibited significant reduction " (lines 129-131) should be statistically supported. There are more than 100 piRNA clusters and most of them are telomeric and pericentric (Brennecke et al 2007). According to the text, the authors did not observe effect of HP1_GLKD on three clusters, 38C,10A and 98F. Thus, the ratio of numbers of telomeric/pericentromeric to euchromatic clusters should be compared to the level of their piRNA loss in different piRNA pathway mutants and HP1_KD. Perhaps, a diagram showing the genomic positions of affected clusters will be helpful.

Fig2c d – Reduction in piRNA abundance, uniquely mapping to transposon insertions in different cytoloocations is shown In Fig. 2 (c,d). It is clear that not all TE insertions are shown in these pictures. How and why these particular insertions were selected? Are these individual copies or TE fragments from different piRNA clusters?

Fig 2e: the profiles in this picture are very similar. Probably, HP1 knockdown is just less efficient than ago3 and aub mutant alleles. This result is not convincing .

Lines 138-140. "We examined loss in uniquely mapping total piRNAs to transposon insertions across genome". Which insertions were considered, individual or located within clusters? It is known that only around 20% of individual TEs produce piRNAs (Shpiz et al 2014), while others are silenced at transcriptional level (with assistance of HP1) by the piRNA pathway. Judging from fig S3a, the authors meant the loss of piRNAs produced by piRNA clusters. It should be clearly stated in the text. Moreover, experiments in the section "HP1a-GLKD affects splicing of the piRNA precursors" show that HP1-KD affects processing of the piRNA cluster transcripts.

Lines 181-182. The authors state that "HP1a robustly represses transposon in telomeric regions, whereas not effectively at non-telomeric regions" based on the experiment with HeT-A-lacZ transgenes. However, strong enrichment of these transgenes by HP1 comparable to that at cluster 42AB has been recently reported (Radion et al 2016). HeT-A-lacZ transgenes were shown to be the targets of piRNA-mediated transcriptional silencing. Fig. 3d and Supplementary Fig.4a,b showing modest lacZ upregulation upon HP1_KD just confirm my suspicion that efficiency of HP1_GLKD is not very high in all these experiments.

Lines 266-267 "Our results suggest that HP1a represses transposons predominantly in the regions close to centromeres and telomeres". It is not clear which TEs were analyzed. Are there TEs located within piRNA clusters or are they individual copies?

Lines 301-2. The authors suggest "that HP1a is required for the piRNAs (production?) targeting evolutionarily older transposons for their repression." This phrase is vague. If HP1 is involved in the piRNA production from the long piRNA precursors transcribed by the clusters how is it possible to discriminate between older and recent insertions within the same locus?

Lines 317-319. The authors state that "In contrast to previously postulated downstream to Piwi-piRISC function in transposons repression, our results suggest that HP1a functions upstream to piRNA processing by suppressing piRNA precursor splicing" The authors mixed piRNA targets and piRNA clusters here. It is better to use "In addition to".

Discussion, lines 323-324: the authors cite the article by Marie et al; "it was recently reported that piRNA loss caused by the absence of HP1a in Drosophila embryos is limited to telomeric transposons". They should specify that this relates not to telomeric but subtelomeric repeats. Subtelomeric piRNA clusters should be specified in Fig S2h. Include more information about these data in the Discussion because Marie et al. observed the loss of piRNAs upon HP1 knockdown from subtelomeric but not from 42AB region. The authors should explain the discrepancy between published (Marie et al) and their data demonstrating reduction in the amount of piRNA from 42AB upon HP1GLKD (Fig 3f).

To conclude, the demonstrated role of HP1 in the piRNA production in general is convincing, but its particular role in the piRNA biogenesis from telomeric and pericentric clusters remains unclear. This conclusion should be toned down or supported better. The description of the role of HP1 in TE silencing (or TE piRNA production) is very confusing and should be rewritten in a clearer manner.

Response to Reviewers' comments:

We thank both the reviewers for finding our work useful. We also thank them for providing their concerns, suggestions and critique about our study for constructive review process. We believe that addressing their concerns have improved our study and now supports our conclusions better. We are providing a detailed point-by-point response to their concerns below in italic. To state our responses clearly, we gave the numbers to the points raised by Reviewer #2.

Reviewer #1 (Remarks to the Author):

This work uses a combination of genetic experiments and genomics with bioinformatics to understand the role of HP1a in transposon silencing and piRNA production in *Drosophila*. The authors find that HP1a represses a subset of transposons, predominantly located in telomeric and pericentric regions. This repression is probably due to HP1a-dependent production of piRNAs corresponding to these transposons. This paper suggests that HP1a represses splicing of piRNA precursors derived from telomeric and pericentric regions, thereby ensuring the production of piRNAs corresponding to transposons located in telomeric and pericentric regions. Overall, the paper has relevant information that will be useful for people working on piRNA biogenesis and piRNA-mediated transposon silencing in *Drosophila*. However, comments below indicate aspects that need significant revision before publication. This revision is important to determine the actual impact of the work.

Major comments:

1. This paper lacks experimental data as to how HP1a robustly represses transposons located in telomeric and pericentric regions, whereas not effectively at non-telomeric and non-pericentric regions. Where does the specificity come from?

In this study, we show that HP1a is required for piRNA production from the HP1a-enriched regions (Fig 2d, S3a, 3e and S3a, S4d). The HP1a-GLKD leads to piRNA loss predominantly from telomeric, subtelomeric and regions close to centromeres (Fig 2c, 2d, S3a). HP1a occupancy at these regions is comparatively very high. By contrast, clusters having low HP1a binding did not show piRNA loss in HP1a-GLKD ovaries (Fig 3e and S3d). We believe that HP1a functions for piRNA biogenesis in cis, which is further supported by its involvement in repressing cluster transcript splicing through Rhi (Fig S4e, Fig 4f and g, please see the response below to Point 2). We believe this explains the specificity of HP1a function at the telomeric, subtelomeric and regions close to centromeres.

2. This paper lacks experimental data as to how HP1a represses splicing of piRNA precursors and how HP1a recognizes and regulates splicing of piRNA precursors derived from telomeric and pericentric regions. Where does the specificity come from?

In the revised manuscript, we added new data providing a glimpse of how HP1a represses cluster transcript splicing. We examined the effect on Rhi caused by HP1a-GLDK. The Rhi-Del-Cuff (RDC) complex functions in germline nucleus to license the transcription of piRNA clusters and repress splicing (ZHANG et al. 2014; CHEN et al. 2016). HP1a-GLKD results in significant alteration of Rhi localization (Fig 4f). In addition, HP1a-GLKD causes reduction in Rhi occupancy at clusters closer to telomeric and pericentric regions, such as those on 1A, 20F and 42AB, where HP1a is enriched, while Rhi occupancy at the clusters not enriched with HP1a remained unchanged (Fig. 4g). These data suggest that HP1a functions to support RDC function at the HP1a-enriched regions.

3. Figure 4: Sox102F transcripts are almost completely spliced in control. Thus, it is not a good control to show that HP1a KD does not result in increase in splicing of the transcripts. The authors shall consider analyzing other cluster transcripts.

We thank the reviewer for pointing this out. Unfortunately, it is very difficult to amplify cluster transcripts with a good fidelity, owing to their repetitive nature. To date, only a handful of clusters, at cytolocations 102D, 20F and 42AB and HeT-A antisense transcript have been successfully amplified (ZHANG et al. 2014; AKKOUCHE et al. 2017). In this study, we examined cluster transcript splicing by qRT-PCR from these regions in HP1a-GLKD ovaries (Fig 4a-c).

To further confirm splicing defect, as suggested, we conducted a more robust analysis of our RNA-seq to provide account as splicing for most of the germline clusters (amended Fig 4e). These data more clearly show an increase in precursor splicing from the clusters that exhibited reduction in the piRNAs in HP1a-GLKD ovaries. The cluster transcripts arising from clusters at chromosome X telomere (1A) and other HeT-A regions and those located closer to centromeres, showed significantly high increase in spliced precursors. By contrast, clusters at cytolocations 38C and 20D, 102C, 43C etc did not show any significant change in Rhi occupancy or any significant reduction in piRNAs (Fig. 2c, 4e).

4. It is hard to understand how the data shown in Figure 4b are computed based on the results shown in Figure 4a. Particularly, for 20F. Is the difference (with such large error bars) really significant?

We appreciate the reviewer for raising this point. Data shown in Fig 4a is gel photos of semi-quantitative PCR run on the respective loci. On the other hand, the quantification of Fig 4b is based on the other qPCR using primers detecting solely spliced or

unspliced transcripts (shown in the loci map, green arrows). We amended the legend to state this point more clearly.

“Spliced to unspliced transcript level ratio (SSR) calculated for the piRNA clusters 42AB, 20F, and sox102F. qRT-PCR was conducted with a primer set detecting either unspliced or spliced transcript. Three independent experiments were performed.”

We truly appreciate the reviewer for pointing out the huge error bar for cluster 20F in the previous manuscript. This came from computing standard error calculating absolute expression levels for this cluster. We have now replaced it with the standard deviation of fold change in HP1a-GLKD ovaries replicates, as we did for other clusters. We used Mann-Whitney U test to compare the significance of increase in splicing for all the clusters, including 20F, in Fig 4c. The test shows indeed the upregulation of splicing from 20F is significant. In addition, the gel photos of the semi quantitative PCR clearly showed the enrichment of spliced isoforms compared to the unspliced one.

5. It is difficult to follow Figure 2c and d, Figure 4e, Sup Figure 2h. It is difficult to understand which clusters are located in telomeric and pericentric regions.

We thank the reviewers for pointing this out. Both the coordinate and cytoloactions have been added in the figures. .

6. The paper seems to have been written in a hurry and some aspects of it need to be improved. The paper needs revising in English grammar and syntax

We carefully checked and revised the manuscript for grammar and syntax.

Reviewer #2 (Remarks to the Author):

The study by Teo et al is of interest for different fields including piRNA biology, chromatin structure and gene regulation. This study addresses an important question in the piRNA field, namely the role of HP1 in the piRNA production in the germline. Using whole genome data the authors find that HP1 knockdown in *Drosophila* germline caused substantial decrease in piRNA production by telomeric and pericentric piRNA clusters accompanied by accumulation of spliced piRNA precursors from these loci. Given that HP1 binds not only piRNA targets, but also piRNA cluster regions, it is surprising that the role of HP1 in the piRNA production has not been explored yet. Moreover, this study adds a novel information about an additional role of highly conserved HP1 chromatin protein in the piRNA biogenesis. While the manuscript does not provide mechanistic insights into the particular role of HP1 in the piRNA biogenesis, I find it overall of broad interest. Despite the fact, that the work is well supported by the experimental data, I would suggest a serious revision of manuscript to better present these data, to avoid wrong conclusions and to make the story more convincing.

We thank for reviewer for recognising the significance of our work and for the comments.

1. In Abstract and along all the manuscript, the authors write “loss of HP1” implying not null mutation (which is lethal) but HP1 RNAi knockdown. However, efficiency of different siRNA constructs in knockdown experiments varies highly and is not 100%. Moreover, nanos promoter driving GAL4 expression has specific expression pattern during oogenesis. Thus, these data describe knockdown the efficiency of which is very difficult to estimate because ovaries are a complex tissue containing both somatic and germ cells. It should be taken into consideration and rephrased.

We thank the reviewer for pointing this out. We replaced “loss of HP1a” with “HP1a germline knockdown” or “HP1a depletion in the germline” throughout in the revised manuscript.

As to a possibility of incomplete knocking down by nos-Gal4, we wish to highlight that we used a driver line harbouring NGT40 on the 2nd chromosome and nanos-GAL4 on the 3rd chromosome in our experiments. This GAL4 line drives strong

expression homogeneously across all stages of germline development (GRIEDER et al. 2000). We are providing data on GFP expression driven by three different germline driver lines: nosGAL4, NGT40-GAL4 combined with nos-Gal4 (used in this study) and MTD-GAL4 for comparison. NGT40 with nos-Gal4 driver gave the robust expression of transgene, across all stages of ovarian development, strongly supporting an efficient knockdown by the driver.

2. Results, First section contains observations, which mainly confirm previous results. In Wang and Elgin 2011 strong upregulation of not only HeT-A, but also Blood, Burdock and Bari was observed upon HP1 GLKD (strains from VDRC were used), however, in Tao et al derepression of these TEs was not detected (Fig1b, lanes 71-75). There is also inconsistency in TE derepression data between Fig1b (RNA-seq) and Supplementary Fig1c (RT-PCR).

The difference in transposon derepression patterns upon HP1a-GLKD between our study and Wang and Elgin 2011 may have resulted from fundamentally different RNAi approaches used in our studies. The study by Wang and Elgin (WANG AND ELGIN 2011) used VDRC lines; long hairpin RNAs were driven by NGT40 in conjunction with overexpression of DCR2. In their study, the HP1a-GLKD ovaries were severely atrophic (Fig S7 in Wang and Elgin, 2011). Severely atrophic ovaries may very likely reflect greater magnitude of transposon derepression, and the quantification of transcript in comparison to control may not be very accurate.

The TRiP lines are better suited for the gene knockdown in germline cells, which produce short hairpin RNAs to knockdown genes. Indeed phenotype and piRNA loss upon GLKD of piRNA pathway component Aub and Ago3, with TRiP lines has been shown to mimic that in the mutants (Ni et al. 2011). In our study, we used two different TRiP lines to knockdown HP1a by a strong germline driver, as mentioned above. Both HP1a-GLKD with those RNAi lines showed a similar transposon derepression pattern.

As to differences in expression levels of some transposons between the RNA-seq and qRT-CPR experiments, indeed we observed a slight difference in the fold-change for a few transposons. We used 2-days old flies for RNA-seq and 2-5-days old flies for the qRT-PCRs, which could affect the extent of transposon derepression. Nevertheless, both the RNA-seq and qRT-PCR data showed similar derepression patterns: telomeric transposons showed high derepression while transposon families for e.g I-element, bari etc did not show any significant derepression.

3. Using clonal analysis, the authors confirmed the activation of HeT-A but did not analyze other TEs from Wang and Elgin. I-element is used as a TE which is not activated in HP1_KD. However, it is not the best choice, because I-element is not affected by piwi mutations as well (Klenov et al 2011). Thus,

the conclusion “that HP1a is required in the germline cells to repress a subset of the transposons” is not quite correct.

I-element derepression in piwi mutants has been described in at least two credible studies (CHAMBEYRON et al. 2008; SENTI et al. 2015). In addition, a number of studies showed significant I-element derepression in mutants of many piRNA pathway components including the Piwi-family proteins, Aub and Ago3 (LIM AND KAI 2007; BRENNECKE et al. 2008; MALONE et al. 2009; PATIL AND KAI 2010). Hence, we considered I-element as a transposon that is robustly repressed by piRNA pathway.

As to our conclusion ‘HP1a is required in the germline cells to repress a subset of the transposons’, it is based on derepression of a handful of transposons in HP1a-GLKD ovaries (Fig 1b; Table S2). This conclusion is not solely based on I-element staining in HP1a-GLKD ovaries. Many important transposons such max, bari, invader, 412, mdg1, 17.6, stalker4 and others along with I-element did not show significant derepression in the HP1a-GLKD ovaries (Table 2), while the derepression of these transposons have been reported in many different piRNA pathway mutants. We show that out of 100, 22 transposon families showed derepression in HP1a-GLKD ovaries, these included accord, tirant, copia, mgd3, invader4, and telomeric transposons. These observations led us to conclude about HP1a requirement for repression of subset of transposons.

4. I suppose that efficiency of the HP1_KD was not very high (see below). The authors have to discuss all these discrepancies and soften the conclusion. Actually, only telomeric HeTA seems to be upregulated. This is not surprising because HeT-A is the most sensitive among all TEs to the piRNA pathway disruption. The authors should confirm the conclusion about telomeric TEs upregulation by normalization to the genomic copy numbers. Even in the strains with similar genetic background the copy number of telomeric TEs can be different. Thus, PCR on genomic DNAs of all genotypes used in the study with primers specific to telomeric TEs is essential for normalization of RNA-seq and small RNAseq data.

As mentioned above, in this study, we drove shRNA to knockdown HP1a with a strong germline driver harbouring NGT40 and nanos-Gal4. Nevertheless we could not exclude a possibility of residual function of HP1a. In this revised manuscript, we have argued this possibility in Discussion .

Page 15, line 305; “Although knockdown based approach could very likely reflect milder effect on piRNA biogenesis, due to some residual activity of HP1a, we nevertheless show HP1a function in the piRNA pathway.”

We agree that a robust HeT-A derepression is observed in many piRNA pathway component mutants. However, HP1a-GLKD ovaries showed significant derepression of 17 other canonical transposons, please see the response to previous comment and Fig 1C. We'd also like to point out that HP1a-GLKD showed derepression of copia, which was not observed in the vas mutants (Vagin et al 2004). It suggests shRNA-based knockdown highlights HP1a requirement for repression of several non-telomeric transposons.

As to the possibility of a difference in Het-A copy numbers in the control and HP1a-GLKD strains, we did not observe a significant increase in 15 days old HP1a-GLKD ovaries.

reduction in the piRNAs lead to high HeT-A derepression, but not a dramatic increase in the HeT-A copy number. Therefore, the dramatic HeT-A derepression upon HP1a-GLKD is very likely resulted from loss of repression of HeT-A foci and not because of significant increase in HeT-A duplication in genome.

5. Lines 83-87. Previously, TEs were subdivided into 3 types, germline, somatic and intermediate relative to their ability to be overexpressed upon piwi depletion (Malone et al). The data described here most likely provide confirmation to these results. In Supp Fig1f, expression levels of Idefix and gypsy should be included as a control to the efficiency of HP1_KD in follicular cells. These TEs should be overexpressed because Piwi-mediated transcriptional silencing requires the HP1 (Le Thomas et al., 2013; Rozhkov et al., 2013).

We deeply appreciate the reviewer for raising this issue. Indeed, ZAM is a very good control to gauge efficiency of HP1a_KD in the somatic cells, as its transcriptional repression is most robust among all transposons in the ovarian somatic cells (ROZHKOVA et al. 2013). In addition, many credible studies have shown robust piRNA loss against ZAM upon perturbation of piRNA pathway in somatic cells (MALONE et al. 2009; HAASE et al. 2010; HANDLER et al. 2013; MUERDTER et al. 2013). In revised manuscript, we highlighted this point as below.

Page 6, Line 78; “By contrast, HP1a knockdown in somatic cells uniquely led to repression of ZAM. Piwi-piRNA mediated repression of ZAM is reported to be mediated via HP1a (LE THOMAS et al. 2013; ROZHKOVA et al. 2013).”

6. Fig2b – Both axes are designated as replicate 2, needs correction. I didn't find the numbers of Supplementary Tables neither in file names nor in the Tables themselves.

Thank you for pointing these mistakes. Indeed the Figure 2b compares loss of piRNAs mapping to canonical transposons between two biological replicates of HP1a-GLKD[2] (replicate 1 and 2), and we corrected this. We apologize for not providing a list of supplementary tables, and now we have included it in the revised manuscript.

7. Lines 119-135; description of cluster-derived piRNAs is very confusing mostly due to the absence of their accurate designations and poor pictures. FigS2h and Table S4 show genomic coordinates of the piRNA clusters, while in the text, the authors refer to telomeric, pericentric, 38C, 10A, 42AB and 98F clusters. The readers will not be able to read all these data. The authors should use similar designations for all pictures or at least add the designations of the clusters mentioned in the text to Suppl. Fig2h. For example, in Brennecke et al 2007 the clusters were termed as ML# (master locus #). The phrase “In addition to telomeric clusters, piRNAs mapping to clusters near centromeric regions also exhibited significant reduction “ (lines 129-131) should be statistically supported. There are more than 100 piRNA clusters and most of them are telomeric and pericentric (BRENNKECKE et al. 2007). According to the text, the authors did not observe effect of HP1_GLKD on three clusters, 38C, 10A and 98F. Thus, the ratio of numbers of telomeric/pericentromeric to euchromatic clusters should be compared to the level of their piRNA loss in different piRNA pathway mutants and HP1_KD. Perhaps, a diagram showing the genomic positions of affected clusters will be helpful.

We deeply thank reviewer for raising this issue. In the revised manuscript, the coordinates and cytoloations are mentioned together in Fig S2h (now changed to Fig S2i in revised manuscript). In addition, the clusters are referred by their cytoloations and coordinates in the text. In addition, as suggested, we have included a new figure showing the effect of HP1a-GLKD for all clusters in the chromosomal arms with respect to their genomic location (Fig 2c in the revised manuscript). This figure also includes the coordinated and cytolocation for each cluster. The original Fig 2C has been moved to supplementary figures, as Fig S3c.

As to our following claim, “In addition to telomeric clusters, piRNAs mapping to clusters near centromeric regions also exhibited significant reduction”, we'd like to highlight the genomic locations of clusters affected in the HP1a-GLKD ovaries. HP1a-GLKD led to reduction in piRNAs mapping to 18 clusters. Among these, except cluster at 76D and 42AB, all the clusters were very close to telomeres and centromeres. Generally, the clusters located distal to pericentric regions did not show as severe reduction in piRNAs as observed for clusters closer to centromeres.

In the revised manuscript, we have described this clearly, with the revised Fig2c supporting the above observation. Hence, we did not intend to claim that all the pericentric clusters show significant reduction in HP1a-GLKD ovaries. We'd also like to point out that only 40 out of 100 major clusters are present on the chromosomal arms. We have included the following text in Results of the revised manuscript

Page 7, line 115; "Only 50% of the clusters located on the chromosomal arms, excluding the ones on the heterochromatin and chromosome U, showed reduction in total piRNAs in HP1a-GLKD ovaries. Although the majority of piRNA clusters on chromosomal arms are spread in pericentric and telomeric regions, clusters showing piRNA loss were predominantly very close to telomeres and centromeres (Fig. 2c)."

According to the text, the authors did not observe effect of HP1_{GLKD} on three clusters, 38C, 10A and 98F. Thus, the ratio of numbers of telomeric/pericentromeric to euchromatic clusters should be compared to the level of their piRNA loss in different piRNA pathway mutants and HP1_{KD}.

We wish to highlight that apart from clusters at 38C, 10A and 98F cytoloations, many others at pericentric regions did not show piRNA reduction in HP1a-GLKD ovaries (Fig 2c, s2j in the revised manuscript). To support our claim "In addition to telomeric clusters, piRNAs mapping to clusters near centromeric regions also exhibited significant reduction.", we compared the piRNA loss between the clusters present in cytoloations marking the centromeres and telomeres (1, 20, 21, 40, 41, 60, 61, 80, 81 and 100) with clusters at other cytoloations. We have amended Fig S3blegend to make the above point clear. We have also included the number of clusters in each category.

8. Fig2c d – -Reduction in piRNA abundance, uniquely mapping to transposon insertions in different cytoloations is shown In Fig. 2 (c,d). It is clear that not all TE insertions are shown in these pictures. How and why these particular insertions were selected? Are these individual copies or TE fragments from different piRNA clusters?

Thank you for pointing this out. The figures 2c and 2d represent reduction in the piRNAs mapping to transposon insertions which were consolidated for each cytoloation. In Fig. 2c (now Fig. 2d in the revised version), we only considered the selected cytoloations, where sum of piRNAs uniquely-mapping to transposon insertions exceeded 1000 in the control ovaries in each. In the revised figure 2d, position of each cytoloation is roughly superimposed on chromosomes along with reduction in piRNAs.

To describe it more clearly, we included the following statement in the revised legend of figure 2d;

“Fold reduction in the piRNAs uniquely mapping to transposon insertions, for different cytoloations. The piRNAs uniquely mapping to mapping to transposon insertions were consolidated for each cytoloation and the cytoloations having more than 1000 uniquely-mapping piRNAs in control were analyzed”

We also started with the following sentence while describing the genome-wide changes in transposon mapping piRNAs in the results section.

Page 8, line 140; “This prompted us to examine the genome-wide effect of HP1a-GLKD on piRNAs arising from transposon insertions including those in the piRNA clusters”

9. Fig 2e: the profiles in this picture are very similar. Probably, HP1 knockdown is just less efficient than ago3 and aub mutant alleles. This result is not convincing.

Figure 2e is loess normalization of piRNA loss, which might not sensitively represent the magnitude of piRNA loss between the mutants. The loess normalization of piRNAs loss for chr2 was calculated in 10 kb window. This however, shows that piRNA loss in piRNA pathway mutants is more severe widespread at pericentric regions. Remarkably, piRNA loss in HP1a-GLKD ovaries is as severe as aub and ago3 mutants at many regions. With this analysis, we wanted to convey that piRNA loss resulting from aub and ago3 mutants, is more well spread in pericentric regions compared to HP1a-GLKD ovaries.

10. Lines 138-140. “We examined loss in uniquely mapping total piRNAs to transposon insertions across genome”. Which insertions were considered, individual or located within clusters? It is known that only around 20% of individual TEs produce piRNAs (Shpiz et al 2014), while others are silenced at transcriptional level (with assistance of HP1) by the piRNA pathway. Judging from fig S3a, the authors meant the loss of piRNAs produced by piRNA clusters. It should be clearly stated in the text. Moreover, experiments in the section “HP1a-GLKD affects splicing of the piRNA precursors” show that HP1-KD affects processing of the piRNA cluster transcripts.

As to criteria for including a transposon insertion in the study, we have analyzed the transposon insertions in two different ways. We analyzed them cytoloation wise, including those in piRNA clusters (please see response to Point 8). We also analyzed change in piRNAs for each transposon insertion in fig S3a. For this we examined insertions, again including

those present in piRNA clusters, which had at least 10 reads uniquely mapping to them. This was described in the legend of fig S3a.

To reflect changes in piRNAs mapping to transposon insertion in genome-wide, we amended the text describing above. Please also see the response to point 8.

Page 8, Line 140; “This prompted us to examine the genome-wide effect of HP1a-GLKD on piRNAs arising from transposon insertions including those in the piRNA clusters”

As to the number of insertions that are represented by significant number of piRNAs, we agree with reviewer that majority of individual TE insertions are not targeted by the piRNAs. We found that among 5431 transposons insertions annotated in Flybase, only 2112 are targeted by more than 10 uniquely-mapping piRNAs (23-29 nt in size) in the control small RNA libraries. Out of these, approximately 503 were present in the piRNA clusters. Although majority of piRNAs were mapping to these insertions in piRNA clusters.

11. Lines 181-182. The authors state that “HP1a robustly represses transposon in telomeric regions, whereas not effectively at non-telomeric regions” based on the experiment with HeT-A-lacZ transgenes. However, strong enrichment of these transgenes by HP1 comparable to that at cluster 42AB has been recently reported (Radion et al 2016). HeT-A-lacZ transgenes were shown to be the targets of piRNA-mediated transcriptional silencing. Fig. 3d and Supplementary Fig.4a,b showing modest lacZ upregulation upon HP1_KD just confirm my suspicion that efficiency of HP1_GLKD is not very high in all these experiments.

We thank the reviewer for pointing this out. To address a possibility that inefficient HP1a knockdown resulted in non-robust LacZ expression of Het-A-lacZ, we have performed clonal analysis using a HP1a loss of function allele to examine the expression of HeT-A/LacZ transgene. Approximately 60% of egg-chambers lacking HP1a showed LacZ expression (Fig. 3e in revised manuscript). This observation strengthens observation that HP1a may not function robustly for transposon repression at regions that are not close to telomeres and centromeres.

12. Lines 266-267 “Our results suggest that HP1a represses transposons predominantly in the regions close to centromeres and telomeres”. It is not clear which TEs were analyzed. Are there TEs located within piRNA clusters or are they individual copies?

The above statement summarised the results of small RNA analysis and piRNAs clusters analysis. It includes all the TEs targeted by a significant number of piRNAs, inside or outside

of the piRNA clusters. In the revised version, we added clear descriptions throughout the manuscript. Please kindly refer the response to Point 10 and 12.

13. Lines 301-2. The authors suggest “that HP1a is required for the piRNAs (production?) targeting evolutionarily older transposons for their repression.” This phrase is vague. If HP1 is involved in the piRNA production from the long piRNA precursors transcribed by the clusters how is it possible to discriminate between older and recent insertions within the same locus?

Previous studies suggested that the evolutionarily-old transposons are enriched in the regions close to centromeres (KOFLEER et al. 2012), suggesting that the piRNA clusters close to the centromeres are likely enriched with evolutionarily-old transposons. These clusters are enriched with HP1a and show significant loss in piRNAs upon HP1a-GLKD. We believe HP1a uniquely functions for piRNA biogenesis from such clusters where older transposons predominantly resides. However, it remains elusive whether HP1a differentiate between older and newer transposon insertions at those regions. However, we feel addressing its molecular mechanism is out of the scope of this study. We added our argument about this issue in Discussion.

In Discussion; “The HP1a importance in repression evolutionarily old transposons and their enrichment at pericentric regions might contribute to the compartmentalized function of HP1a in piRNA biogenesis. The highly conserved HP1a may have adapted to silence the evolutionary older transposons in the pericentric and telomeric regions. In contrast, the functions of newly-evolved and Drosophila-specific proteins such as Rhi and Cuff in the maintenance of the cluster transcripts genome-wide may indicate the adaptation of the piRNA pathway to novel transposon threats and reproductive isolation.”

14. Lines 317-319. The authors state that “In contrast to previously postulated downstream to Piwi-piRISC function in transposons repression, our results suggest that HP1a functions upstream to piRNA processing by suppressing piRNA precursor splicing” The authors mixed piRNA targets and piRNA clusters here. It is better to use “In addition to”.

We thank for the reviewer for suggestion. Indeed our study suggests that HP1a works downstream to piRNA processing for repression of selected transposons. For transposon tirant, accord and transib1, which are derepressed upon the HP1a-GLKD, we did not observe any changes in total cellular piRNAs, while we observed reduction in the Piwi-bound piRNAs mapping to them. In the revised manuscript, we describe this in Results (Fig S2) and in Discussion as below;

Page 7, line 102; “As an exception to the trend observed for other transposons, upregulation of accord, tirant and transib1, was accompanied by the loss of the Piwi-

bound piRNAs, but the levels of total cellular piRNAs targeting these transposons were not reduced (Supplementary Fig. 2g)."

In Discussion; "Our results also suggest that HP1a functions downstream to piRNA processing for repression of several transposons, such as tirant, accord, and transib1."

15. Discussion, lines 323-324: the authors cite the article by Marie et al; "it was recently reported that piRNA loss caused by the absence of HP1a in Drosophila embryos is limited to telomeric transposons". They should specify that this relates not to telomeric but subtelomeric repeats. Subtelomeric piRNA clusters should be specified in Fig S2h. Include more information about these data in the Discussion because Marie et al. observed the loss of piRNAs upon HP1 knockdown from subtelomeric but not from 42AB region. The authors should explain the discrepancy between published (Marie et al) and their data demonstrating reduction in the amount of piRNA from 42AB upon HP1GLKD (Fig 3f).

We appreciate the reviewer for raising these issues. As rightly pointed out, Ronsseray and colleagues (MARIE et al. 2016) reported loss of piRNAs mapping to telomeric transposons HeT-A, TART and TAHRE. In addition, they described severe reduction in the piRNAs from cluster at 3R-tip (as described in MARIE et al. 2016). Consistent with their observation in embryonic stages, HP1a-GLKD in adult ovaries caused most severe piRNA reduction from this cluster at 100E (3R-tip) (Fig 2c, d). Hence, both the studies highlight HP1a importance in piRNA biogenesis from telomeres.

However, as pointed out by the reviewer, they did not report any change in the piRNAs mapping to mapping to 42AB in the embryos. This difference may be because of gonadal stages. The piRNA populations and pathway function differs between the embryos and adult ovaries. In the embryos, the piRNA pathway functions to establish the epigenetic state of the clusters and ping-pong amplification with the help of maternally inherited piRNAs (BRENNECKE et al. 2008; MALONE et al. 2009; AKKOUCHE et al. 2017). Nevertheless their study during embryonic stages also supports HP1a requirement in piRNA production from telomeres in their case. We added a statement in Discussion to highlight the observation in embryo as below.

Page 15, Line 311; "Ronsseray and colleagues reported that HP1a-GLKD in Drosophila embryos causes piRNAs loss against telomeric transposons and cluster at 3R-tip(MARIE et al. 2016). Like the embryos, the cluster at 3R-tip, designated as cluster at 100E, consistently shows most severe piRNA loss among all clusters, upon HP1a-GLKD in the adult germline as well (Fig 2c). However, they did not observe reduction in piRNAs mapping to cluster at 42AB or other regions. In the embryonic stages the epigenetic state of the clusters and ping-pong amplification are established, while the adult gonads have a

completely functional piRNA pathway^{26, 36, 44}. This might lead to differences in observation about HP1a function in piRNA pathway between different development stages.”

16. To conclude, the demonstrated role of HP1 in the piRNA production in general is convincing, but its particular role in the piRNA biogenesis from telomeric and pericentric clusters remains unclear. This conclusion should be toned down or supported better. The description of the role of HP1 in TE silencing (or TE piRNA production) is very confusing and should be rewritten in a clearer manner.

We thank the reviewer for pointing this out. We indeed have replaced the conclusions in Results at many places with the observations. In addition, we hope that we present convincing lines of evidence in the revised manuscript, claiming that the piRNA loss in the HP1a-GLKD ovaries was predominantly coming from regions close to centromeres and telomeres, where HP1a is highly enriched. Newly added clonal analysis of HP1a with HeT-A/LacZ transgene further supports that HP1a function in transposon silencing is more robust at telomeric regions. HP1a involvement in suppressing the cluster transcript splicing predominantly from the clusters close to telomeric and centromeric regions explains its specificity. In the revised manuscript, we also provided the effect on location of Rhi and its occupancy upon HP1a-GLKD, suggesting that this splicing defect is likely caused through Rhi function.

In the revised manuscript, we present a more elaborate and clear description of reduction pattern of cluster-mapping and genome-wide transposon insertion mapping piRNAs as described above.

Altogether, we now hope that this supports more convincingly HP1a requirement for the piRNA production from specific genomic regions.

References:

- Akkouche, A., B. Mugat, B. Barckmann, C. Varela-Chavez, B. Li *et al.*, 2017 Piwi Is Required during Drosophila Embryogenesis to License Dual-Strand piRNA Clusters for Transposon Repression in Adult Ovaries. *Mol Cell* 66: 411-419 e414.
- Brennecke, J., A. A. Aravin, A. Stark, M. Dus, M. Kellis *et al.*, 2007 Discrete small RNA-generating loci as master regulators of transposon activity in Drosophila. *Cell* 128: 1089-1103.
- Brennecke, J., C. D. Malone, A. A. Aravin, R. Sachidanandam, A. Stark *et al.*, 2008 An epigenetic role for maternally inherited piRNAs in transposon silencing. *Science* 322: 1387-1392.

- Chambeyron, S., A. Popkova, G. Payen-Groschene, C. Brun, D. Laouini *et al.*, 2008 piRNA-mediated nuclear accumulation of retrotransposon transcripts in the *Drosophila* female germline. *Proc Natl Acad Sci U S A* 105: 14964-14969.
- Chen, Y. A., E. Stuwe, Y. Luo, M. Ninova, A. Le Thomas *et al.*, 2016 Cutoff Suppresses RNA Polymerase II Termination to Ensure Expression of piRNA Precursors. *Mol Cell* 63: 97-109.
- Grieder, N. C., M. de Cuevas and A. C. Spradling, 2000 The fusome organizes the microtubule network during oocyte differentiation in *Drosophila*. *Development* 127: 4253-4264.
- Haase, A. D., S. Fenoglio, F. Muerdter, P. M. Guzzardo, B. Czech *et al.*, 2010 Probing the initiation and effector phases of the somatic piRNA pathway in *Drosophila*. *Genes Dev* 24: 2499-2504.
- Handler, D., K. Meixner, M. Pizka, K. Lauss, C. Schmied *et al.*, 2013 The genetic makeup of the *Drosophila* piRNA pathway. *Mol Cell* 50: 762-777.
- Kofler, R., A. J. Betancourt and C. Schlotterer, 2012 Sequencing of pooled DNA samples (Pool-Seq) uncovers complex dynamics of transposable element insertions in *Drosophila melanogaster*. *PLoS Genet* 8: e1002487.
- Le Thomas, A., A. K. Rogers, A. Webster, G. K. Marinov, S. E. Liao *et al.*, 2013 Piwi induces piRNA-guided transcriptional silencing and establishment of a repressive chromatin state. *Genes Dev* 27: 390-399.
- Lim, A. K., and T. Kai, 2007 Unique germ-line organelle, nuage, functions to repress selfish genetic elements in *Drosophila melanogaster*. *Proc Natl Acad Sci U S A* 104: 6714-6719.
- Malone, C. D., J. Brennecke, M. Dus, A. Stark, W. R. McCombie *et al.*, 2009 Specialized piRNA pathways act in germline and somatic tissues of the *Drosophila* ovary. *Cell* 137: 522-535.
- Marie, P. P., S. Ronsseray and A. Boivin, 2016 From Embryo to Adult: piRNA-Mediated Silencing Throughout Germline Development in *Drosophila*. G3 (Bethesda).
- Muerdter, F., P. M. Guzzardo, J. Gillis, Y. Luo, Y. Yu *et al.*, 2013 A genome-wide RNAi screen draws a genetic framework for transposon control and primary piRNA biogenesis in *Drosophila*. *Mol Cell* 50: 736-748.
- Ni, J. Q., R. Zhou, B. Czech, L. P. Liu, L. Holderbaum *et al.*, 2011 A genome-scale shRNA resource for transgenic RNAi in *Drosophila*. *Nat Methods* 8: 405-407.
- Patil, V. S., and T. Kai, 2010 Repression of Retroelements in *Drosophila* Germline via piRNA Pathway by the Tudor Domain Protein Tejas. *Curr Biol*.
- Rozhkov, N. V., M. Hammell and G. J. Hannon, 2013 Multiple roles for Piwi in silencing *Drosophila* transposons. *Genes Dev* 27: 400-412.
- Senti, K. A., D. Jurczak, R. Sachidanandam and J. Brennecke, 2015 piRNA-guided slicing of transposon transcripts enforces their transcriptional silencing via specifying the nuclear piRNA repertoire. *Genes Dev* 29: 1747-1762.
- Wang, S. H., and S. C. Elgin, 2011 *Drosophila* Piwi functions downstream of piRNA production mediating a chromatin-based transposon silencing mechanism in female germ line. *Proceedings of the National Academy of Sciences of the United States of America* 108: 21164-21169.
- Zhang, Z., J. Wang, N. Schultz, F. Zhang, S. S. Parhad *et al.*, 2014 The HP1 homolog rhino anchors a nuclear complex that suppresses piRNA precursor splicing. *Cell* 157: 1353-1363.

Reviewers' comments:

Reviewer #1 (Remarks to the Author):

On the whole it has been improved and the authors have addressed many of the reviewers concerns. It is an important contribution to our understanding of the piRNA biogenesis in *Drosophila*.

However, the aspect that is the least satisfying with this work is that we are still left with essentially mechanisms for how HP1a regulates the splicing of transcripts arising from telomeres and centromeres and how the inhibition of the splicing leads to piRNA production. Why and how are spliced piRNA precursors resistant to the piRNA production machinery?

Reviewer #2 (Remarks to the Author):

I highly appreciate the effort the authors put in the revision of the manuscript. I found the figures that describe piRNAs mapping to the piRNA clusters (Fig 2) much more comprehensible. In the revised version, the authors included experiments showing Rhino delocalization upon HP1 knockdown (Fig.4f). This fact is interesting and confirms the idea that HP1 acts upstream of piRNA processing.

However, I still have some comments to the manuscript.

1. Lines 90-92, add reference
2. The protein symbols along the text should be written in capital letters (Piwi, Rhi, Cuff etc).
3. Lines 111-113: "To investigate whether HP1a represses transposons in the ovarian somatic cells as well, we knocked down HP1a expression in somatic cells using both RNAi lines". TE derepression upon HP1_KD in ovarian somatic cells was already studied by several groups (Ohtani et al 2013, Sienski et al 2015). Strong activation of Zam and gypsy was observed in OSC upon HP1_KD.
Lines 116-117: The statement "HP1a knockdown in somatic cells led to derepression of transposons, which are targeted by piRNAs in the somatic cells including ZAM" does not correspond to the data. In Fig.S1f, no Zam activation is observed upon HP1_tjGAL4_KD. The authors' explanations that different knockdown systems cause different effects on TE expression (in "Responses to comments") seem to be not satisfactory in this case because Zam is a well confirmed robust target of the piRNA-mediated transcriptional silencing. I would suggest to remove Fig S1f and paragraph 111-118.
4. Fig 2C Y axis is not titled
5. Fig S2j is not mentioned in the text.
6. I still believe that the experiment with HeT-A-lacZ adds nothing to the story. Fig 3C shows wrong proportion of HeT-A and lacZ. It is not correct to compare silencing mechanisms of 400bp fragment of the HeT-A promoter and of endogenous telomeric arrays. HP1 is equally enriched at 42AB and HeT-A-lacZ transgene, however, HeT-A-lacZ is not a dual-strand piRNA cluster (Radion et al 2017). Thus, the conclusion that this transgene is not as effectively silenced by HP1 as telomeric HeT-A arrays is debatable. The statement in the lines 214-215 that "all egg chambers exhibited endogenous HeT-A accumulation (Fig. 3c and 3d)" does not agree with the lower panel of Fig.3c. Actually, HeT-A expression is observed here only in one egg chamber (6 stage). In addition, I am surprised by the complete coincidence of the beta-gal and HeT-A GAG staining in the oocyte. I would suggest moving this experiment to Supplementary materials.
7. The paragraph including lines 366-375 should be rewritten or removed. It contains incorrect conclusion that transgenes inserted in subtelomeric region are silenced due to their telomeric

position. Actually, any sequences inserted within piRNA cluster become its integral part and this is the fundamental principle of adaptivity (Muerder et al 2012, Khurana et al 2011). Reference (50) is irrelevant.

8. In Fig 5b and Fig S6a, Y axis is designated as Fold reduction (\log_{10}). I guess it should be \log_2 or FC.

9. HP1 ChIP-seq was done on ovaries which consist of somatic and germline cells. So, correlation between piRNA level in the HP1_GLKD and ovarian HP1 ChIP is not quite correct and might be used as an indirect evidence. It should be noted in the text and toned down in the Abstract.

10. Check spelling, e.g. duplication "mapping to mapping to" appears along the manuscript, figure legends and axis titles.

11. Headlines are still not added to Tables.

Response to Reviewers' comments:

We would like to thank the editor and the reviewers for this constructive review process. Indeed, inputs from reviewers tremendously contributed to improve our manuscript. We have read comments from the reviewers in this round, and are submitting a revised manuscript in which all changes are highlighted as requested by the editor. We also provide a point-by-point response in blue italic to the reviewers' comments below. We believe we have satisfactorily addressed their concerns and that manuscript now very clearly shows novel role of HP1a in piRNAs precursor maturation for genome defense in germline cells.

In addition to reviewer 2 comments, we would like to amend our analysis of the piRNAs mapping to evolutionarily old transposons in HP1a-GLKD ovaries. In light of earlier comments from reviewers and consulting some experts in the field, we realize that we should omit some transposon insertions whose evolutionary age has not yet been established robustly^{1,2}. Hence we have excluded previous Supplementary figure 6c and Line 111-118 in the previous manuscript. In addition, for the same reasons we have also dropped few transposon insertions from our analysis, shown in figure 5b.

We will like to stress that these exclusion do not change our results (Fig 5b, supplementary Fig 6b, c). On the contrary, our claim that HP1a knockdown leads to greater loss in piRNAs targeting evolutionarily old transposons is better supported. The statistical tests for revised dataset in Fig 5b support our claim more robustly.

Point-by-point Response

Reviewer #1 (Remarks to the Author):

On the whole it has been improved and the authors have addressed many of the reviewers concerns. It is an important contribution to our understanding of the piRNA biogenesis in *Drosophila*.

However, the aspect that is the least satisfying with this work is that we are still left with essentially mechanisms for how HP1a regulates the splicing of transcripts arising from telomeres and centromeres and how the inhibition of the splicing leads to piRNA production. Why and how are spliced piRNA precursors resistant to the piRNA production machinery?

We thank the reviewer for recognizing the contribution of our study.

As rightly pointed out, it remains elusive how upregulation of the splicing contributes to impediment in the piRNA processing. Many credible published studies undertaking above point as center of their investigation could not show the mechanistic details. Current studies including ours have built the foundation for future studies for addressing mechanistic details.

We show that HP1a loss causes perturbation of Rhi occupancy, which leads to increase in piRNA precursor splicing. We think that we have provided lines of evidence for HP1a function in piRNA biogenesis upstream to Piwi-piRISC mediated silencing, giving an impact to the field.

Reviewer #2 (Remarks to the Author):

I highly appreciate the effort the authors put in the revision of the manuscript. I found the figures that describe piRNAs mapping to the piRNA clusters (Fig 2) much more comprehensible. In the revised version, the authors included experiments showing Rhino delocalization upon HP1 knockdown (Fig.4f). This fact is interesting and confirms the idea that HP1 acts upstream of piRNA processing.

We thank the reviewer for appreciating that revised manuscript confirms HP1a function upstream of piRNA processing.

However, I still have some comments to the manuscript.

1. Lines 90-92, add reference

Thanks for pointing this out. We added the reference.

2. The protein symbols along the text should be written in capital letters (Piwi, Rhi, Cuff etc).

We amended carefully.

3. Lines 111-113: “To investigate whether HP1a represses transposons in the ovarian somatic cells as well, we knocked down HP1a expression in somatic cells using both RNAi lines”.

TE derepression upon HP1_KD in ovarian somatic cells was already studied by several groups (Ohtani et al 2013, Sienski et al 2015). Strong activation of Zam and gypsy was observed in OSC upon HP1_KD.

Lines 116-117: The statement “HP1a knockdown in somatic cells led to derepression of transposons, which are targeted by piRNAs in the somatic cells including ZAM” does not correspond to the data. In Fig.S1f, no Zam activation is observed upon HP1_tjGAL4_KD. The authors` explanations that different knockdown systems cause different effects on TE expression (in “Responses to comments”)

seem to be not satisfactory in this case because Zam is a well confirmed robust target of the piRNA-mediated transcriptional silencing. I would suggest to remove Fig S1f and paragraph 111-118.

We thank the reviewer pointed this out. The ZAM primers used in the study were designed to amplify a region at 3'end. We have now re-examined ZAM expression using a different set of primers, which amplifies a region in the middle of ZAM transcript (reported in ³). These primers show increased upregulation of ZAM upon HP1a-GLKD in the ovarian somatic cells (figure appended to this response). Although ZAM derepression was not as robust as it was shown upon HP1a-GLKD in the OSC cells in the abovementioned studies we think it could be because of difference in the state of cells i.e. in vivo and culture.

Nevertheless, we have now deleted this section, as suggested by reviewer. This is a control experiment in somatic cells, which is not relevant to our study about germline function of HP1a.

4. Fig 2C Y axis is not titled

We thank for reviewer for pointing this out. The label was obscured during PDF conversion. We have amended the figure.

5. Fig S2j is not mentioned in the text.

We thank reviewer for pointing out this mistake. We have wrongly cited it as figure 2i. We have correctly cited the figure in revised manuscript.

6. I still believe that the experiment with HeT-A-lacZ adds nothing to the story. Fig 3C shows wrong proportion of HeT-A and lacZ. It is not correct to compare silencing mechanisms of 400bp fragment of the HeT-A promoter and of endogenous telomeric arrays. HP1 is equally enriched at 42AB and HeT-A-lacZ transgene, however, HeT-A-lacZ is not a dual-strand piRNA cluster (Radion et al 2017). Thus, the conclusion that this transgene is not as effectively silenced by HP1 as telomeric HeT-A arrays is debatable. The statement in the lines 214-215 that “all egg chambers exhibited endogenous HeT-A accumulation (Fig. 3c and 3d)” does not agree with the lower panel of Fig.3c. Actually, HeT-A expression is observed here only in one egg chamber (6 stage). In addition, I am surprised by the complete coincidence of the beta-gal and HeT-A GAG staining in the oocyte. I would suggest moving this experiment to Supplementary materials.

We appreciate reviewer's comments. We would like to highlight that experiments conducted with HeT-A-LacZ transgene was performed to examine whether HP1a functions more robustly in piRNA-mediated repression downstream to piRNA processing at telomeric regions. We think that HP1a enrichment at telomeric regions supports observations from HeT-A/LacZ experiment. To clearly represent this intention, we have amended the sentence at beginning of section that describes the HeT-A lacZ experiment.

“To investigate if HP1a function is required for piRNA-mediated repression of the transposons in location-dependent manner, we examined the effects of HP1a-GLKD on the repression of HeT-A located at non-telomeric regions”

We also amended the conclusion in this section to reflect above clearly.

“further supporting that HP1a functions more robustly for piRNA-mediated transposon repression at telomeric regions”

In addition, we have also amended Discussion to better convey our claim that HP1a functions more robustly for piRNA-mediated downstream repression at telomeric regions (covered in the response to next point).

We did not intend to compare any similarity between the mechanisms of LacZ transgene repression with that for telomeric HeT-A array. We recognize that latter are a distinct class of piRNA cluster, which act as both the source of piRNAs and their targets, while the former is target of silencing by piRNAs originating from cluster. Remarkably, we found an increase in the antisense transcription and increase in splicing of antisense transcripts from telomeric HeT-A cluster in HP1a-GLKD ovaries, indicating an upstream role of HP1a for piRNA biogenesis from telomeric HeT-A arrays (Fig 3e and supplementary Fig 5a).

We cannot completely rule out the possibility that continued silencing of LacZ transgene in many HP1a-GLKD ovarioles is due to incomplete knockdown. We speculate that the manifestation of incomplete silencing could be attributed to activity of residual maternal LacZ-HeT-A piRNAs, which could engage the downstream components such as Silencio.

As pointed out, in Fig 3C, one optical section imaged by a confocal microscope do not show the HeT-A expression in many egg chambers. The z-projection of all optical sections shows HeT-A expression in other egg chambers. However, z-projection highlight HP1a in other issue, i.e., HP1a-positive follicle cells mask HP1a-negative germline cells. For the reviewer's reference, we provide the z-projected image of the previous Fig 3C below, showing robust Het-A signals in more egg chambers (arrows denote Het-A positive germarium and egg chambers).

In the revised manuscript, we replaced the previous image with the other ovariole where LacZ expression in all egg-chambers could be represented by z-projection of fewer optical sections, excluding ones with follicle cells.

Regarding to LacZ signal, as pointed out, we also noticed the signal of anti-LacZ antibody for HetA-LacZ showed up in the oocyte, like endogenous Het-A signal. According to the study reported this transgene, this transgene contains few amino acids from HeT-A protein at N-terminal of beta Gal. We suspect this could be the reason.

7. The paragraph including lines 366-375 should be rewritten or removed. It contains incorrect conclusion that transgenes inserted in subtelomeric region are silenced due to their telomeric position. Actually, any sequences inserted within piRNA cluster become its integral part and this is the fundamental principle of adaptivity (Muerder et al 2012, Khurana et al 2011). Reference (50) is irrelevant.

We agree with the reviewer. We now amended this part by removing a part of it and the reference, as below.

“Genetic analysis using HeT-A/lacZ reporter further demonstrated that HP1a-role in downstream piRNA-mediated transposon repression is more robust at the telomeric regions.”

8. In Fig 5b and Fig S6a, Y axis is designated as Fold reduction (log10). I guess it should be log2 or FC.

We truly appreciate the reviewer for pointing this out. We amended this mistake.

9. HP1 ChIP-seq was done on ovaries which consist of somatic and germline cells. So, correlation between piRNA level in the HP1_GLKD and ovarian HP1 ChIP is not quite correct and might be used as an indirect evidence. It should be noted in the text and toned down in the Abstract.

We thank the reviewer for reminding this. HP1a ChIP was performed using the total ovarian lysates. Hence, we have amended the relevant parts throughout the manuscript.

10. Check spelling, e.g. duplication “mapping to mapping to” appears along the manuscript, figure legends and axis titles.

We amended those.

11. Headlines are still not added to Tables.

We added those in the revised manuscript.

1. Petrov DA, Fiston-Lavier AS, Lipatov M, Lenkov K, Gonzalez J. Population genomics of transposable elements in *Drosophila melanogaster*. *Mol Biol Evol* **28**, 1633-1644 (2011).
2. Kofler R, Betancourt AJ, Schlotterer C. Sequencing of pooled DNA samples (Pool-Seq) uncovers complex dynamics of transposable element insertions in *Drosophila melanogaster*. *PLoS Genet* **8**, e1002487 (2012).
3. Guida V, *et al.* Production of Small Noncoding RNAs from the flamenco Locus Is Regulated by the gypsy Retrotransposon of *Drosophila melanogaster*. *Genetics* **204**, 631-644 (2016).

REVIEWERS' COMMENTS:

Reviewer #1 (Remarks to the Author):

In this twice reviewed manuscript, the authors have responded thoroughly to all critiques.

Reviewer #2 (Remarks to the Author):

I am satisfied how the authors responded to the comments. There is just one more thing I would like to ask (which would not need re-review). The suggestion that "LacZ derepression in significant number of HP1a-GLKD ovarioles may have been mediated by the residual or maternally-deposited HeT-A/lacZ piRNAs" is not quite correct because this transgene does not produce piRNAs complementary to lacZ (Radion et al 2018) and is regulated by the telomeric HeT-A piRNAs. Thus, the paragraph (lines 370-373) should be rephrased to avoid unnecessary speculations: Genetic analysis using HeT-A/lacZ reporter further demonstrated that the role of HP1a in the piRNA-mediated transposon repression is more robust at the telomeric regions whereas lacZ derepression is observed only in a limited number of HP1a-GLKD ovarioles.